# Insulin-activated store-operated Ca$^{2+}$ entry via Orai1 induces podocyte actin remodeling and causes proteinuria

Ji-Hee Kim[1,2,3], Kyu-Hee Hwang[1,2,3], Bao T. N. Dang[1,2,3], Minseob Eom[4], In Deok Kong[1,2], Yousang Gwack [5], Seyoung Yu[6], Heon Yung Gee [6], Lutz Birnbaumer[7,8], Kyu-Sang Park [1,2,3] & Seung-Kuy Cha [1,2,3✉]

Podocyte, the gatekeeper of the glomerular filtration barrier, is a primary target for growth factor and Ca$^{2+}$ signaling whose perturbation leads to proteinuria. However, the effects of insulin action on store-operated Ca$^{2+}$ entry (SOCE) in podocytes remain unknown. Here, we demonstrated that insulin stimulates SOCE by VAMP2-dependent Orai1 trafficking to the plasma membrane. Insulin-activated SOCE triggers actin remodeling and transepithelial albumin leakage via the Ca$^{2+}$-calcineurin pathway in podocytes. Transgenic *Orai1* over-expression in mice causes podocyte fusion and impaired glomerular filtration barrier. Conversely, podocyte-specific *Orai1* deletion prevents insulin-stimulated SOCE, synaptopodin depletion, and proteinuria. Podocyte injury and albuminuria coincide with Orai1 upregulation at the hyperinsulinemic stage in diabetic (*db/db*) mice, which can be ameliorated by the suppression of Orai1-calcineurin signaling. Our results suggest that tightly balanced insulin action targeting podocyte Orai1 is critical for maintaining filter integrity, which provides novel perspectives on therapeutic strategies for proteinuric diseases, including diabetic nephropathy.

---

[1] Department of Physiology, Yonsei University Wonju College of Medicine, Wonju, Republic of Korea. [2] Department of Global Medical Science, Yonsei University Wonju College of Medicine, Wonju, Republic of Korea. [3] Mitohormesis Research Center, Yonsei University Wonju College of Medicine, Wonju, Republic of Korea. [4] Department of Pathology, Yonsei University Wonju College of Medicine, Wonju, Republic of Korea. [5] Department of Physiology, David Geffen School of Medicine, University of California, Los Angeles, CA 90095, USA. [6] Department of Pharmacology, Graduate School of Medical Science, Brain Korea 21 Project, Yonsei University College of Medicine, Seoul, Republic of Korea. [7] Neurobiology Laboratory, National Institute of Environmental Health Sciences, Research Triangle Park, NC 27709, USA. [8] Institute of Biomedical Research (BIOMED), School of Medical Sciences, Catholic University of Argentina, C1107AAZ Buenos Aires, Argentina. ✉email: skcha@yonsei.ac.kr

Podocytes are highly specialized visceral epithelial cells found in the kidney glomerulus and function as gatekeepers in the glomerular filtration barrier. A proper actin cytoskeleton in podocytes is critical for maintaining the filter integrity; thus, cytoskeletal rearrangement plays an important pathogenic role in proteinuric diseases[1,2]. Close regulation of intracellular $Ca^{2+}$ ($[Ca^{2+}]_i$) is crucial for regulating actin dynamics in podocytes, and aberrant $[Ca^{2+}]_i$ signaling has been postulated as an early event in glomerular dysfunction, including podocyte injury and proteinuria[3,4]. $Ca^{2+}$ influx in non-excitable cells is mainly mediated by the class C transient receptor potential (TRPC) and the store-operated $Ca^{2+}$ (SOC) channels[3,5]. Hyperactivation of the TRPC6 or TRPC5 channel in podocytes induces actin remodeling and causes proteinuria, while inhibition of either channel protects barrier function[3,4,6–9]. Although SOCE is a major $Ca^{2+}$ influx mechanism in epithelial cells, the molecular components and pathophysiological role of SOCE in podocytes remain unclear.

Podocytes respond to external stimuli and convey outside-in signals via membrane receptors, including receptor tyrosine kinases (RTKs), to regulate multiple cellular functions[10]. Deregulation of RTK signaling contributes to the podocyte cytoskeletal rearrangement causing proteinuria[10–14]. Among podocyte growth factors targeting RTKs, insulin is an important factor associated with cell growth and survival[15]. Since insulin receptor signaling is vital for podocyte function[13,16], perturbation of insulin signaling impairs the glomerular filtration barrier causing proteinuria[11–13,17–19]. Genetic deletion of the insulin receptor or its downstream effector phosphoinositide-3-kinase (PI3K) in mouse podocytes aggravates albuminuria[13,17]. However, contrary to its protective actions, insulin is also known to accelerate albuminuria and glomerulosclerosis[11,12]. Acute infusion of insulin elicits transient albuminuria under euglycemic conditions in humans[18]. These studies demonstrate that insulin receptor signaling has an ambivalent action on podocytes' function in maintaining the kidney filter integrity.

Deregulation of $[Ca^{2+}]_i$ or RTK signaling has been implicated in the dysfunction of podocyte slit diaphragm[3,10]. RTK activation can regulate SOCE and alter $[Ca^{2+}]_i$ homeostasis in various tissues[20–23]. To date, the molecular components of SOCE in podocytes and its pathophysiological role in glomerular filter function have not been investigated. Here, we identified that Orai1-mediated SOCE, a critical $Ca^{2+}$ influx pathway, is a downstream effector of insulin signaling that affects podocyte actin dynamics and albumin permeabilities. While Orai1 activation by acute insulin exposure causes reversible actin remodeling and slit-diaphragm plasticity, chronic insulin stimulation of Orai1 leads to irreversible abnormalities causing filter barrier disruption and pathologic proteinuria. Furthermore, this demonstrates a requirement of a fine-tuned insulin signaling; an aberrant insulin signaling may threaten podocyte $[Ca^{2+}]_i$ homeostasis and glomerular functions.

## Results

### Orai1 and STIM1 are the functional molecular components of SOCE in mouse podocytes.
In non-excitable epithelial cells, the major $Ca^{2+}$ influx is mediated by TRPC and SOC channels that are activated by phospholipase C (PLC)-linked Gαq-coupled receptors or RTKs[5,24]. The exaggerated $Ca^{2+}$ signaling through TRPC5/6 has been implicated in podocyte injury[3,6–9,25,26]. Therefore, we examined whether SOCE is another functional $Ca^{2+}$ entry mechanism in isolated glomeruli and podocytes in addition to TRPC5 and TRPC6. Functional SOCE after ER $Ca^{2+}$ depletion was recorded in isolated glomeruli (Fig. 1a–c) and cultured mouse podocytes (Fig. 1d, e), which were blocked using

an Orai1/3 inhibitor, namely GSK-7975A, or a pan-SOCE inhibitor, namely 2-APB. In the case of ruptured whole-cell patch-clamp recordings, ER $Ca^{2+}$ depletion triggered $Ca^{2+}$ release-activated $Ca^{2+}$ (CRAC) channel current showing characteristic inwardly rectifying cation currents in mouse podocytes (Fig. 1f). CRAC channels are composed of pore-forming subunits (Orai1–3) and ER $Ca^{2+}$ sensors (STIM1/2)[5]. We observed that Orai1 was highly expressed in cultured mouse podocytes (Fig. 1g). Orai1 silencing using siRNA revealed that this isoform is the major pore-forming subunit of SOCE in mouse podocytes (Fig. 1h–j). In the case of depleted ER $Ca^{2+}$ pool, the canonical SOCE component STIMs are oligomerized and translocated to the plasma membrane, thereby triggering $Ca^{2+}$ influx through plasmalemmal Orai channels[5]. We observed that cultured podocytes expressed both STIM1 and 2, but knockdown of STIM1 (but not STIM2) markedly inhibited SOCE in podocytes (Fig. 1k–m).

TRPC6 channel is one of the $Ca^{2+}$ entry mechanisms in podocytes[3] and TRPC channels have been suggested to contribute to SOCE in other types of cells[24]. We examined whether TRPC6 participates in SOCE. SOCE was not altered in TRPC6-silenced mouse podocytes (Supplementary Fig. 1a–c) or isolated glomeruli from $Trpc6$ knockout ($Trpc6^{-/-}$) mice (Supplementary Fig. 1d, e). Together, these results demonstrate that SOCE in mouse podocytes is mediated by Orai1 with STIM1, but not by TRPC6.

### Insulin upregulates Orai1-mediated SOCE aggravating actin remodeling and transepithelial albumin leakage.
Podocyte is the main target of growth factors whose deregulation contributes to podocyte dysfunction leading to proteinuria[11,12]. However, SOCE regulation by growth factors and its downstream consequences in podocytes remains unclear. First, we demonstrated that multiple growth factors including serum, insulin or insulin-like growth factor 1 (IGF1) can stimulate SOCE in mouse podocytes (Fig. 2a). We focused on insulin signaling since it is critical for podocyte function[11–13,17]. Insulin treatment increased Orai1-mediated SOCE in isolated glomeruli (Fig. 2b, c) and podocytes (Fig. 2d–f) in a dose-dependent manner (Fig. 2g). It has been reported that TRPC6 is activated by insulin in cultured podocytes[27]. However, insulin-stimulated SOCE was not attenuated in either the TRPC6-silenced cultured podocytes (Supplementary Fig. 1f, g) or the isolated glomeruli from $Trpc6^{-/-}$ mice (Supplementary Fig. 1h, i), demonstrating that TRPC6 may not be involved in SOCE activation by insulin.

Maintaining the actin structure in podocyte foot processes is vital for glomerular filtration[1,28]. Alterations in podocyte actin dynamics can disturb the glomerular filter integrity. For instance, exaggerated $[Ca^{2+}]_i$ signaling alters actin dynamics causing podocyte dysfunction and albuminuria[3,8,29]. Synaptopodin, a substrate of the $Ca^{2+}$-activated phosphatase, calcineurin, is an essential regulator of the podocyte actin cytoskeleton, which is downregulated in chronic proteinuric diseases[28,30]. Next, we monitored whether insulin-activated Orai1 induces actin remodeling associated with synaptopodin degradation via the $Ca^{2+}$-calcineurin pathway. We observed that insulin induced the cortical distribution of actin filaments (Fig. 2h, i) and synaptopodin depletion (Fig. 2j–m). Additionally, insulin resulted in decreased focal adhesions and increased the motility of podocytes (Supplementary Fig 2a–d). All of these alterations were ameliorated by inhibiting Orai1 using siRNA-mediated knockdown or pharmacological blockers (GSK or 2-APB) and by stabilizing synaptopodin using cyclosporine A (CsA), a calcineurin inhibitor[28,30] (Fig. 2h–m and Supplementary Fig 2a–d). Deregulation of $Ca^{2+}$ signaling induces pathological foot process effacement, thereby causing albumin leakage. Consistent with

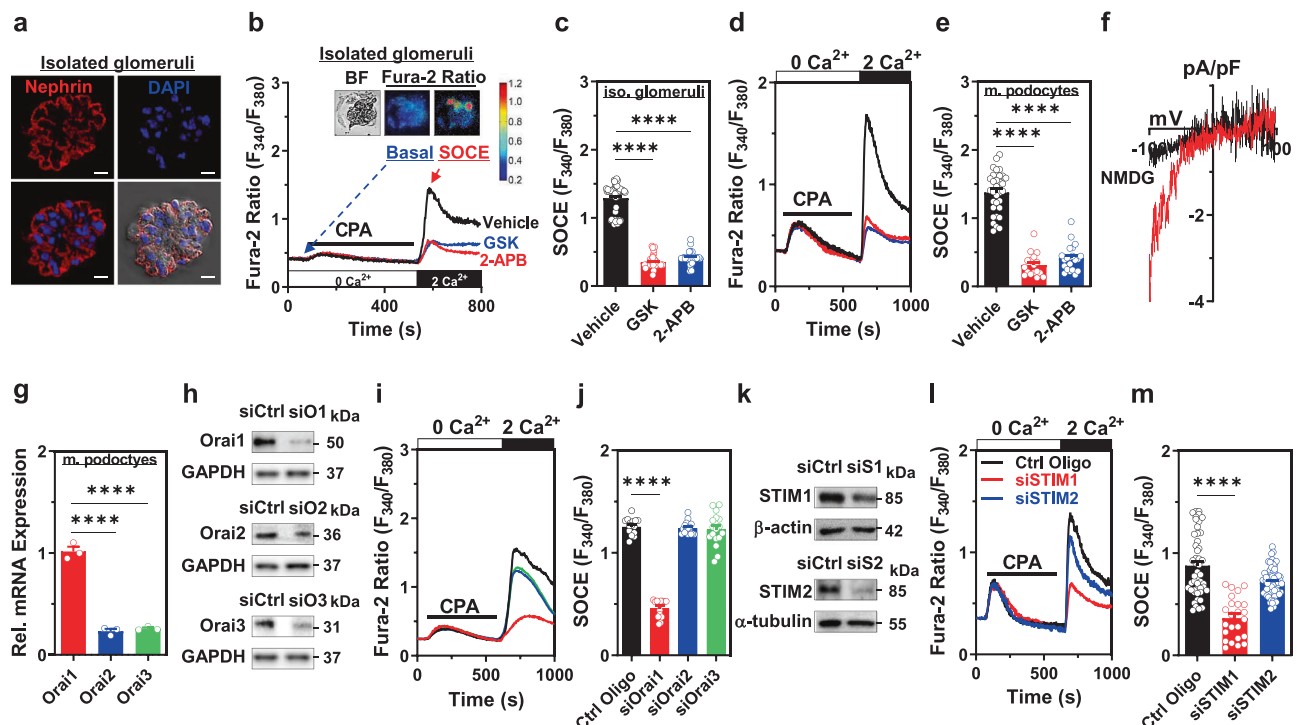

**Fig. 1 Orai1/STIM1 is a primary molecular component of functional store-operated Ca²⁺ entry (SOCE) in podocytes. a** Immunofluorescence images of nephrin (red) and DAPI (blue) stains in isolated mouse glomeruli and differential interference contrast (DIC) results combined with those of nephrin and DAPI staining. Scale bar = 10 μm. **b** Effect of Orai inhibitors on functional SOCE in isolated mouse glomeruli. Representative images of bright-field (BF) and fura-2 fluorescence microscopy of podocytes in the basal and SOCE states, carried out in situ on acutely isolated mouse glomeruli, in the same manner as described in the previous reports[9]. Pre-treatment with Orai inhibitors, GSK (GSK-7975A; 3 μM) or 2-APB (100 μM), was performed 1 h before measuring SOCE. **c** Summary of the SOCE in **b**. $n = 51$ (WT), 39 (GSK), and 33 (2-APB) isolated (iso.) glomeruli per group. **d** Effects of Orai inhibitors on SOCE in immortalized cultured mouse podocytes. **e** Summary of the SOCE in **d**. $n = 32$ (WT), 17 (GSK), and 21 (2-APB) cells per each group. **f** Current–voltage ($I$–$V$) relationship of the CRAC current in cultured mouse podocytes. **g** Relative mRNA levels of Orai channels (1, 2, and 3) in mouse podocytes. $n = 3$ independent experiments. **h** Validation of the siRNA knockdown of the Orai isoforms in mouse podocytes. Control Oligo (siCtrl) was used as non-targeting control siRNA and GAPDH was used as a loading control. **i** Effect of the siRNA knockdown of Orai proteins on SOCE in mouse podocytes. **j** Summary of the SOCE in **i**. $n = 16$ (Ctrl Oligo), 14 (siOrai1), 16 (siOrai2), 17 (siOrai3) cells per each group. **k** Validation of siRNA knockdown of STIM1 and 2 in mouse podocytes. **l** Effect of siRNA silencing of STIM1 or 2 on SOCE in mouse podocytes. **m** Summary of the SOCE in **l**. $n = 48$ (Ctrl Oligo), 24 (siSTIM1), 67 (siSTIM2) cells per each group. All experiments were repeated three times independently (**a, h, k**). Bar graphs are expressed as mean ± SEM. One-way ANOVA followed by Dunnett's multiple comparisons test (**c, e, g, j**, and **m**). ****$p < 0.0001$.

Orai1-induced actin remodeling, insulin elicited podocyte retraction (Fig. 2n) and augmented transepithelial albumin fluxes (Fig. 2o), all of which were reversed by the inhibition of Orai1 or calcineurin (Fig. 2n, o). This supports the notion that Orai1 upregulation by insulin causes podocyte cytoskeletal remodeling and albumin leakage by aggravating Ca²⁺-calcineurin-mediated synaptopodin degradation.

**Podocyte-specific *Orai1* deletion abrogates insulin-stimulated SOCE and albuminuria in mice.** To explore whether podocyte Orai1 is responsible for the insulin-triggered impairment of podocyte filter, we generated podocyte-specific *Orai1*-deletion mice by crossing floxed *Orai1* mice with podocyte-specific Cre recombinase mice driven by podocin promoter/enhancer region (Supplementary Fig. 3a, b). Transcriptional levels of *Orai1* were markedly decreased in the isolated glomeruli of podocyte-specific *Orai1*-deletion (Cre+;*fl/fl*) mice compared with the glomeruli of the wild-type or heterozygous (Cre+;*fl/+*) mice, whereas *Orai1* expression in the medullary fraction was not significantly different between the groups (Fig. 3a). Consistent with these results, we confirmed that Orai1 protein level was significantly reduced in the isolated glomeruli from the Cre+;*fl/fl* mice (Fig. 3b). Immunofluorescence staining with a podocyte-specific marker, synaptopodin, also showed the attenuation of Orai1 expression in the

podocytes of Cre+;*fl/fl* mice (Fig. 3c). Podocyte-specific *Orai1*-deletion mice were viable and demonstrated no apparent changes in kidney size (Supplementary Fig. 3c) and renal histology (Fig. 3d). Functionally, native SOCE in isolated glomeruli was significantly reduced in podocyte-specific *Orai1*-deletion mice when compared to the wild-type or heterozygous mice (Fig. 3e, f) demonstrating that Orai1 is the main component of SOCE in podocytes in vivo.

Next, we demonstrated the exclusive role of Orai1 in insulin-triggered SOCE, synaptopodin dissolution, and albuminuria in vivo. Podocyte-specific *Orai1*-deletion abrogated insulin-activated SOCE in isolated glomeruli (Fig. 3g, h). Insulin-induced albuminuria (Fig. 3i) and synaptopodin degradation (Fig. 3j, k) were markedly reduced by podocyte-specific *Orai1* deletion in mice. These data strongly support that podocyte Orai1 is a crucial target for insulin's detrimental actions on the glomerular barrier. Genetic ablation of TRPC6 (*Trpc6⁻/⁻*) did not affect insulin-induced albuminuria in mice (Supplementary Fig. 4), suggesting that insulin signaling targeting podocyte Orai1 but not TRPC6 is critical for albuminuria.

**Albuminuria caused by Orai1 overexpression in mice is further aggravated by insulin.** To further support the exclusive role of Orai1 in insulin-induced podocyte dysfunction, we employed an

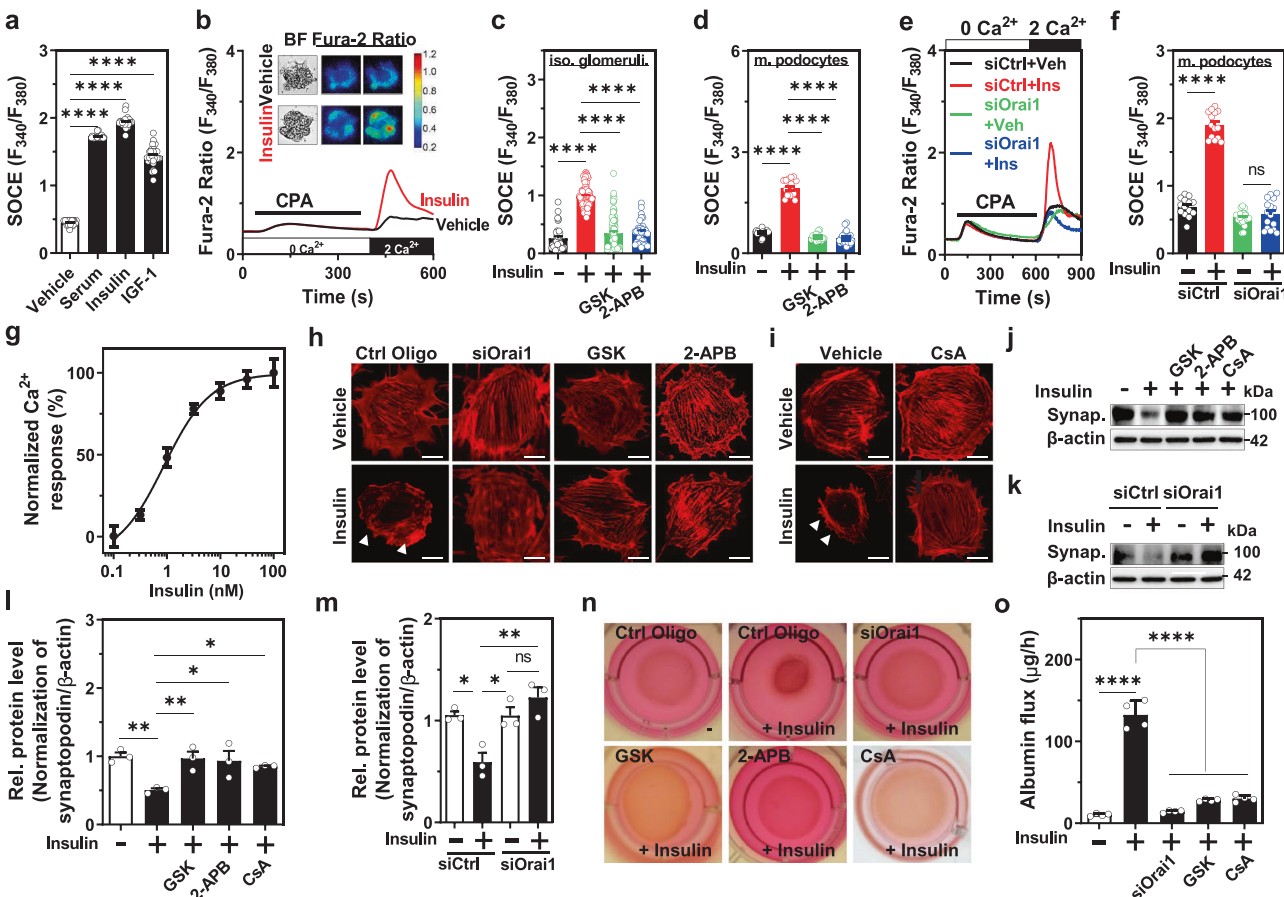

**Fig. 2 Insulin stimulates Orai1-mediated store-operated Ca$^{2+}$ entry (SOCE), aggravating actin rearrangement and transepithelial albumin leakage in mouse podocytes. a** Vehicle ($n = 29$ cells) or serum growth factors including serum (10%) ($n = 26$ cells), insulin (100 nM for 1 h, $n = 27$ cells), and IGF-1 (10 nM for 1 h, $n = 29$ cells) increased SOCE in podocytes. To examine insulin effects, cells or isolated glomeruli were incubated in a serum-free medium for ~16 h before insulin treatment for the indicated duration. **b** Representative SOCE traces with the images of bright-field and fura-2 fluorescence (upper panel) microscopy of insulin-treated podocytes (100 nM) carried out in situ on acutely isolated mouse glomeruli. **c, d** Effect of SOCE inhibitors [GSK (GSK-7975A; 3 μM) or 2-APB (100 μM)] on insulin-stimulated SOCE in isolated (iso.) glomeruli (**c**) and mouse (m.) podocytes (**d**). $n = 33$ (Vehicle), 77 (Insulin), 74 (Insulin+GSK), and 33 (Insulin+2-APB) glomeruli (**c**) and 19 (Vehicle), 17 (Insulin), 15 (Insulin+GSK), and 21 (Insulin+2-APB) cells (**d**) per group. **e** Representative SOCE traces showing the effect of Orai1 knockdown by siRNA on insulin-stimulated SOCE in cultured mouse podocytes. **f** Summary of the SOCE in **e**. $n = 14$ (siCtrl+Veh), 14 (siCtrl+Insulin), 20 (siOrai1+Veh), and 18 (siOrai1+Insulin) cells per each group. **g** Dose–response of insulin on SOCE in cultured mouse podocytes. $n = 21, 21, 32, 10, 33, 15, 19$ cells for each point, respectively. EC$_{50}$ of insulin is approximately 0.95 nM. **h, i** Effect of Orai1-mediated SOCE inhibition by siRNA Orai1 (siOrai1) or GSK or 2-APB (**h**) and effect of cyclosporine A (CsA; 10 μM) (**i**) on insulin-mediated actin rearrangement (white arrow). Scale bar = 50 μm. The experiments were repeated twice and several cells were analyzed for each experiment. **j** Effect of SOCE inhibitors and CsA on insulin-mediated depletion of synaptopodin (Synap.) in cultured mouse podocytes. **k** Effect of siRNA Orai1 on the insulin-induced dissolution of synaptopodin. **l, m**. Summary of relative synaptopodin expression in the **j** and **k**, respectively. $n = 3$ independent experiments. **n** Collagen contraction assay showing insulin-stimulated podocyte retraction, which is rescued by the inhibition of Orai1-calcineurin pathway. This assay was repeated two times independently. **o** Effect of Orai1 knockdown and GSK and CsA treatment on insulin-mediated transepithelial albumin leakage in mouse podocytes. $n = 4$ independent experiments. Bar graphs are expressed as mean ± SEM and analyzed by one-way ANOVA followed by Tukey's multiple comparisons test. *$p < 0.05$, **$p < 0.01$, ***$p < 0.001$, ****$p < 0.0001$; ns not significant (**a, c, d, f, l, m, o, p**).

Orai1-overexpressing mouse model by tail vein injection of *Orai1* transgene. After gene delivery, recombinant mCherry-Flag-tagged Orai1 protein was detected in the kidney cortex extracts (Fig. 3l). Additionally, a substantial portion of mCherry fluorescence was co-localized with synaptopodin in confocal imaging (Fig. 3m), confirming the overexpression of Orai1 in podocytes. Compared to the control, mice overexpressing *Orai1* transgene developed significant albuminuria (Fig. 3n) indicating that excessive activation of Orai1 can induce slit-diaphragm dysfunction and albumin leakage. Intraperitoneal (i.p.) injection of insulin to Orai1-overexpressing mouse markedly increased albuminuria (Fig. 3n). Furthermore, Orai1 overexpression caused synaptopodin depletion (Fig. 3o, p) and foot process effacement (Fig. 3q) in mice, which were aggravated by insulin administration. These results

suggest that the gain of Orai1 function itself is sufficient to impair podocyte slit diaphragm and filtration barrier, which could be further deteriorated by insulin action targeting podocyte Orai1.

**Insulin increases cell-surface abundance of Orai1 via PI3K-dependent VAMP2-associated channel exocytosis.** Next, we examined how insulin activates Orai1-induced SOCE. Orai1 activation by insulin occurs via the alteration of intrinsic channel properties and/or channel trafficking. First, we evaluated Orai1 channel properties and its regulation through insulin treatment. To determine the precise kinetics, CRAC current was measured in a HEK293FT cell line heterologously expressing *Orai1* with *STIM1* to prevent the development of space-clamp error and overcompensation due to the big membrane capacitance of

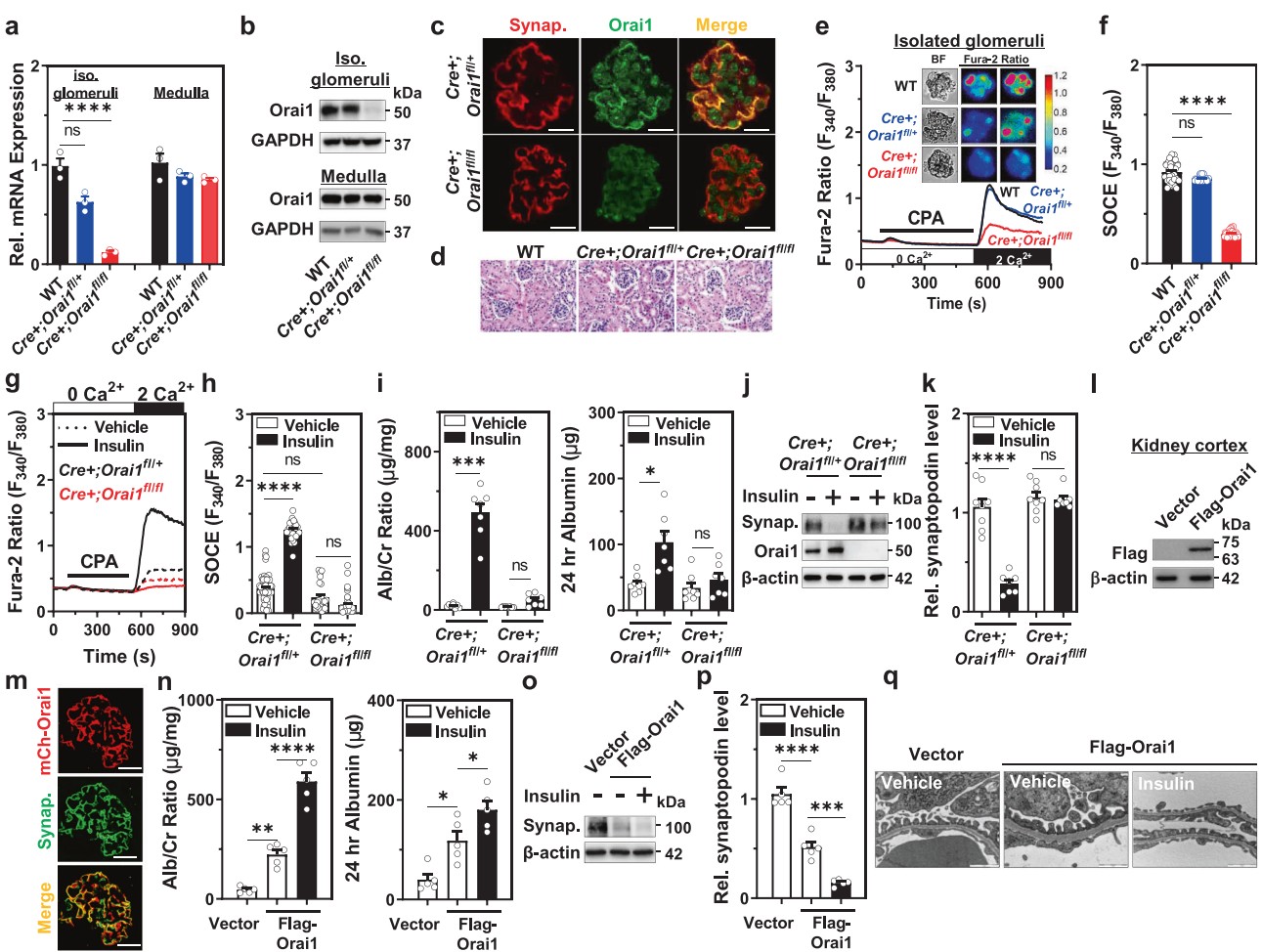

**Fig. 3 Insulin activation targeting the Orai1 channel induces albuminuria in mice. a, b** mRNA (**a**) and protein (**b**) expression levels of Orai1 in the isolated glomeruli and medulla of wild-type (WT), *Cre+;Orai1$^{fl/+}$*, and *Cre+;Orai1$^{fl/fl}$* mice. $n = 3$ independent experiments. **c** Immunofluorescence image displaying Orai1 (green) and synaptopodin (red; Synap.) expression in the isolated glomeruli of *Cre+;Orai1$^{fl/+}$* and *Cre+;Orai1$^{fl/fl}$* mice. Scale bar = 25 μm. All images were selected among 3 mice in each group by independent experiments. **d** H & E staining of the kidney sections of the WT, *Cre+;Orai1$^{fl/+}$*, and *Cre+;Orai1$^{fl/fl}$* mice. **e** Representative traces of SOCE with the images of bright-field and fura-2 fluorescence (upper panel) microscopy carried out in situ on acutely isolated mouse glomeruli from WT, *Cre+;Orai1$^{fl/+}$*, and *Cre+;Orai1$^{fl/fl}$* mice. **f** Quantification of SOCE in **e**. $n = 30$ (WT), 25 (*Cre+;Orai1$^{fl/+}$*), and 29 (*Cre+;Orai1$^{fl/fl}$*) glomeruli per group. **g** Representative traces of fura-2 ratio showing insulin effects on SOCE in acutely isolated mouse glomeruli from *Cre+;Orai1$^{fl/+}$* and *Cre+;Orai1$^{fl/fl}$* mice. **h** Quantification of SOCE in **g**. $n = 60, 26, 24,$ and 48 glomeruli per each group, respectively. **i** Effects of insulin administration (i.p.; 5 U/kg) on 24 h urinary albumin/creatinine ratio (Alb/Cr Ratio) (left) and albumin excretion (right) in *Cre+;Orai1$^{fl/+}$* and *Cre+;Orai1$^{fl/fl}$* mice. $n = 8, 7, 8,$ and 7 mice per group. **j** Immunoblotting analysis showing insulin effects on synaptopodin in isolated glomeruli from *Cre+;Orai1$^{fl/+}$* and *Cre+;Orai1$^{fl/fl}$* mice. **k** Densitometry of relative (Rel.) synaptopodin expression in **j**. $n = 8, 7, 8,$ and 7 mice per each group. **l** Immunoblotting analysis showing the expression of exogenous Orai1 protein (detected by anti-Flag antibody) 48 h after in vivo gene delivery of Flag-tagged-Orai1. **m** Immunostaining of mCherry-tagged Orai1 (red; mCh-Orai1) and synaptopodin (green; Synap.) in mice injected with *Orai1* transgene. Scale bar = 25 μm. Immunostaining was repeated two times from 3 mice each. **n** Effect of transgenic *Orai1* overexpression and insulin administration (*i.p.*; 5 U/kg) on urinary Alb/Cr ratio (left) and albumin excretion. $n = 5$ mice each. **o** Immunoblotting analysis showing the effect of transgenic *Orai1* overexpression and insulin inoculation (i.p.; 5 U/kg) on synaptopodin expression. **p** Densitometry of relative (Rel.) synaptopodin levels in **o**. $n = 5$ mice each. **q** Representative TEM images of foot processes from in vivo gene delivery of Flag-tagged-Orai1 with/without administration of insulin. EM repeated in 3 mice each. Magnification is ×30,000. Scale bar = 1 μm. Bar graphs are expressed as mean ± SEM and analyzed by two-way ANOVA followed by Tukey's multiple comparisons test. *$p < 0.05$, **$p < 0.01$, ***$p < 0.001$, ****$p < 0.0001$; ns not significant (**a, f, h, i, k, n, p**).

podocytes[8]. Consistent with Ca$^{2+}$ imaging experiments, insulin increased the amplitude of the Orai1-mediated CRAC current (Fig. 4a, b). However, scaled-up relative Ca$^{2+}$ influx kinetics and current–voltage (*I–V*) relationship curves were not affected by insulin (see insets of Fig. 4a, c) implying that insulin does not alter intrinsic Orai1 channel properties.

An alternative explanation is that insulin increases CRAC current by promoting the cell-surface abundance of the Orai1 channel. Growth factors promote the insertion of channels into the plasma membrane via the PI3K-dependent pathways[8,31,32]. In a time kinetic analysis, SOCE activation was detected after a 10-min incubation of insulin and reached maximal stimulation at 1 h (Fig. 4d), demonstrating the rapid upregulation of Orai1 by insulin. Consistent with these results, insulin increased the cell-surface abundance of the Orai1 channel within 10 min of incubation without altering the total endogenous Orai1 protein level in cultured podocytes (Fig. 4e, f). This was also confirmed in Orai1-overexpressing HEK293FT cells (Supplementary Fig. 5a–c). Vesicular exocytosis is mediated by SNARE proteins such as VAMP2. The disruption of the SNARE complex by cleaving

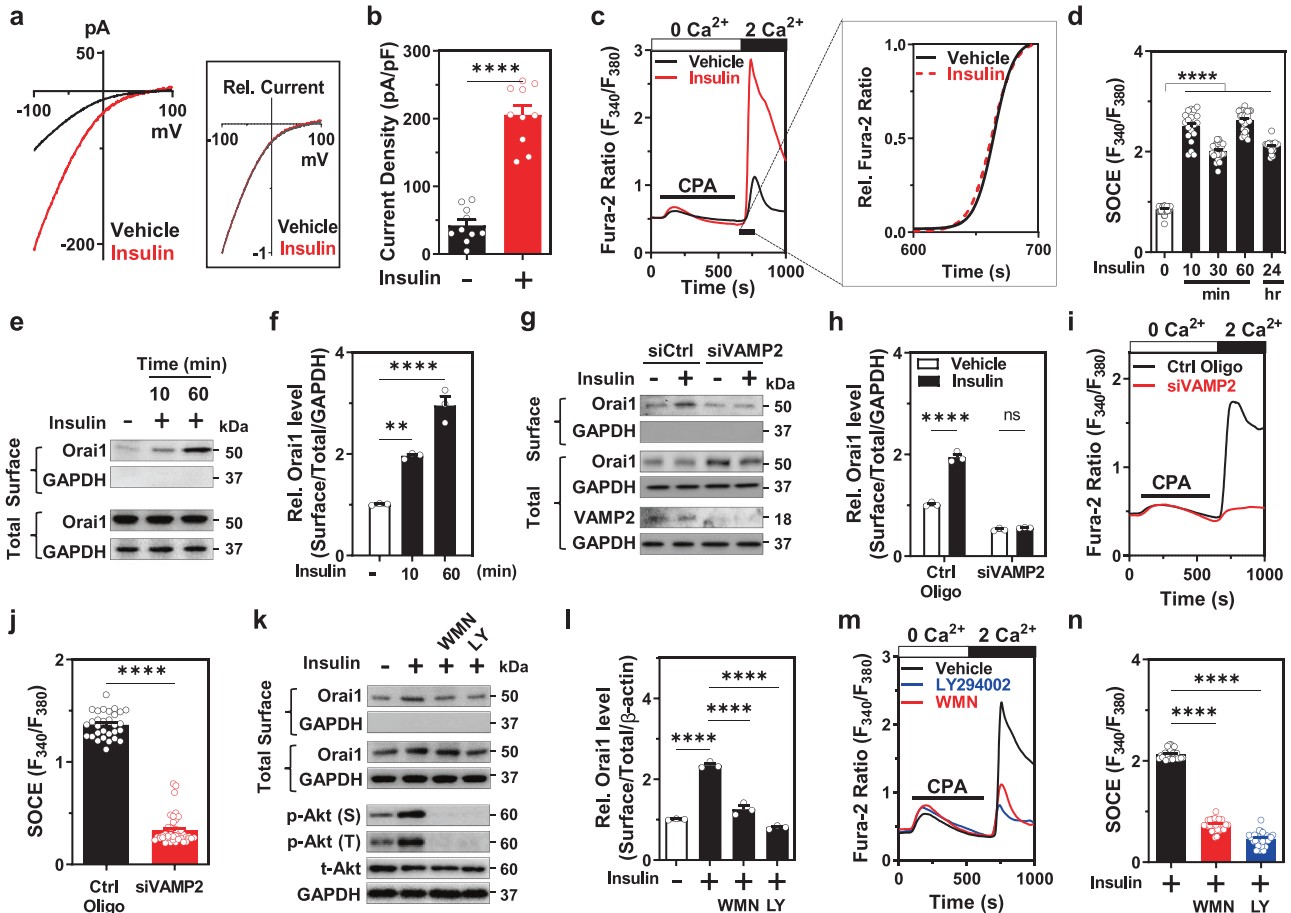

**Fig. 4 Insulin stimulates cell-surface abundance of Orai1 via VAMP2-associated exocytosis in a PI3K-dependent pathway. a** Effect of insulin on the CRAC current in HEK293FT cells heterologously co-expressing 3xflag-mCherry-tagged Orai1 and YFP-tagged STIM1. Inset shows current–voltage relationship of the relative (Rel.) CRAC current with or without insulin treatment. **b** Summary of the Orai1 current density at −100 mV in **a**. $n = 10$ for Vehicle and Insulin-treated cells. **c** Representative traces of SOCE in cultured mouse podocytes from mice treated with vehicle (HCl) or insulin (100 nM). Inset shows the slope of the scaled-up relative $Ca^{2+}$ rise kinetics between 600 and 700 s. **d** Time-window of insulin stimulation on store-operated $Ca^{2+}$ entry (SOCE) in podocytes. $n = 28, 20, 21, 27$, and 19 cells, respectively. **e** Biotinylation assay showing the effect of insulin (100 nM, 10 min, or 1 h) on the cell-surface abundance of Orai1. **f** Quantification of the surface Orai1 by insulin in **e**. $n = 3$ independent experiments. **g** Effect of the knockdown of VAMP2 by siRNA on insulin-stimulated Orai1 cell-surface expression in podocytes. **h** Quantification of the results in **g**. $n = 3$ independent experiments. **i** Representative fura-2 ratio showing the effect of VAMP2 siRNA knockdown on insulin-stimulated SOCE. **j** Summary of the SOCE in **i**. $n = 30$ and 43 cells for Ctrl Oligo and siVAMP2, respectively. **k** Biotinylation assay representing the effects of PI3K inhibitors (wortmannin, WMN; 100 nM and LY294002, LY; 10 μM, treatment for 1 h) on the insulin-stimulated cell-surface abundance of Orai1 in cultured mouse podocytes. Lower panel: immunoblotting analysis validating PI3K-Akt pathway response to insulin stimulation (100 nM, 15 min). Akt phosphorylation at serine 473 (p-Akt (S)) and at threonine 308 (p-Akt (T)). **l** Quantification of the **k**. $n = 3$ independent experiments. **m** Representative SOCE traces showing the effect of PI3K inhibitors on insulin-induced SOCE. **n** Summary of the SOCE in **m**. $n = 28, 29$, and 33 cells per each group. Data are expressed as mean ± SEM and analyzed by one-way ANOVA followed by Dunnett's (**d**, **f**, **n**) or Tukey's (**h**, **l**) multiple comparisons test, and unpaired two-tailed Student's $t$-test (**b**, **j**). ****$p < 0.0001$; ns not significant.

VAMP2 using tetanus toxin (TeNT) impairs SOCE[33,34]. To support the notion that insulin promotes SOCE via exocytosis of the channel, we found that insulin-induced increase in SOCE and cell-surface abundance of Orai1 was blocked by pre-treatment with brefeldin A (BFA) or TeNT to disrupt vesicular exocytosis (Supplementary Fig. 6a–f). Similarly, VAMP2 knockdown also attenuated the potentiating effects of insulin (Fig. 4g–j). Together, our results support that insulin stimulates SNARE-associated vesicular insertion of Orai1 into the plasma membrane of the podocyte.

The PI3K-Akt pathway is the major downstream signaling cascade of the insulin receptor in podocytes[13]. We previously reported that serum increases the cell-surface trafficking of Orai1 through the stimulation of the PI3K-driven exocytosis of the channel[35]. Here, we demonstrated that insulin increased Akt phosphorylation at serine and threonine residues and also

increased the surface abundance of Orai1 in mouse podocytes; these effects of insulin action were prevented by PI3K inhibitors (Fig. 4k, l). Consistent with these results, PI3K inhibitors also reduced insulin-stimulated SOCE (Fig. 4m, n). Collectively, these data suggest that insulin promotes the exocytosis of Orai1-carrying vesicles via the activation of the PI3K pathway.

**Orai1 is overexpressed in the early phase of hyperinsulinemic *db/db* mice.** The above results support the notion that the acute activation of Orai1 by insulin stimulation induces transient disorganization of the cytoskeletal structure in podocytes resulting in glomerular filter disruption and albuminuria. In contrast to acute Orai1 activation by insulin, long-term incubation of insulin upregulated SOCE and Orai1 expression in podocytes (Supplementary Fig. 7a–d). Therefore, we examined the effects of long-

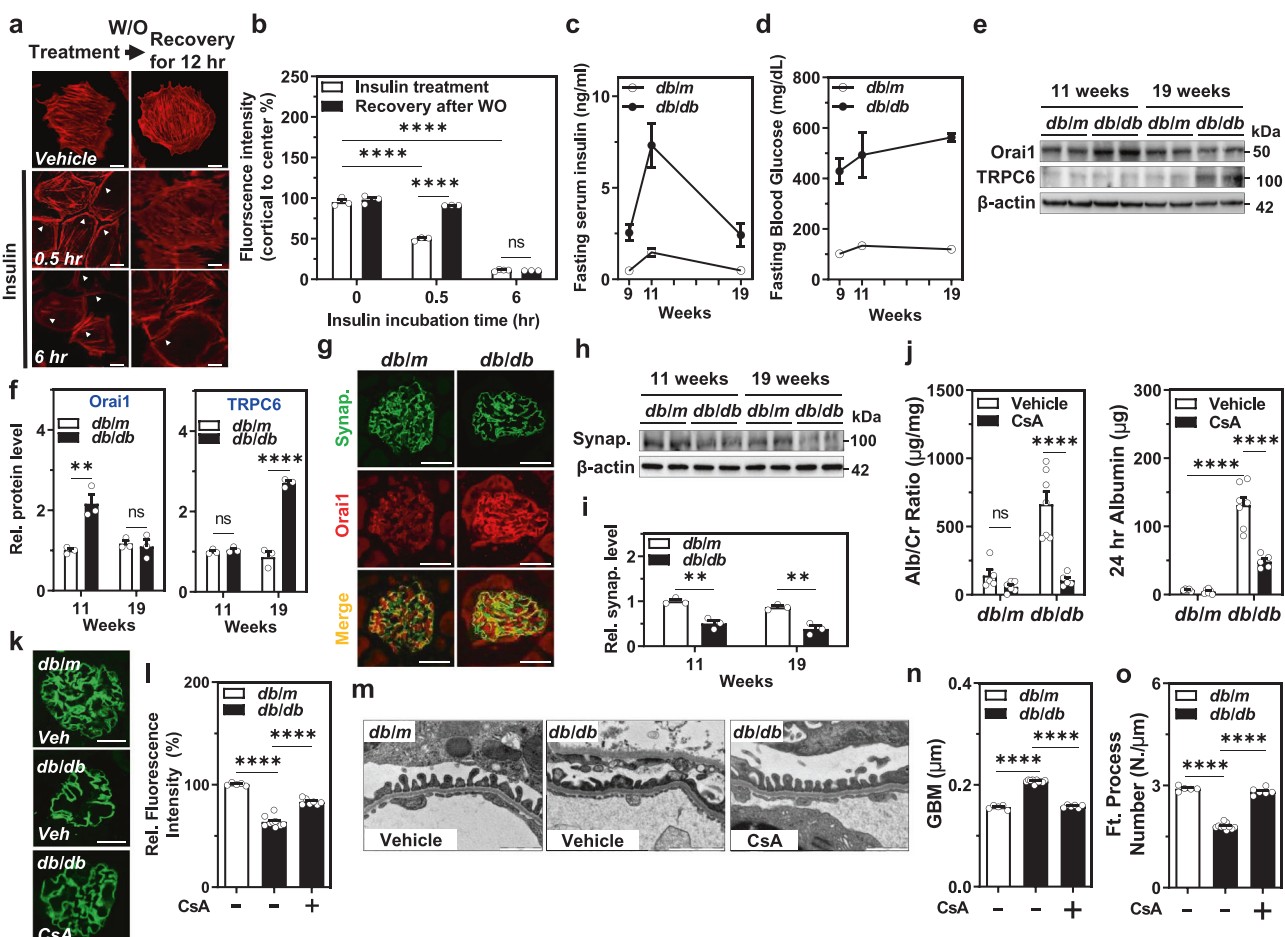

**Fig. 5 Orai1 is overexpressed in hyperinsulinemic *db/db* mice and CsA ameliorates slit-diaphragm disruption and albuminuria. a** Time-dependent recovery of insulin-mediated actin rearrangement. Podocytes were incubated with insulin for 30 min (short-term) or 6 h (long-term), washed out (W/O), and further incubated in an insulin-free medium for 12 h before phalloidin staining. White arrows indicate actin rearrangement by insulin treatment. Scale bar = 50 μm. **b** Quantification of the actin fiber intensity in **a**. $n$ = 3, 3, 4, 5, 4, and 3 dishes, twenty images in each dish. **c, d** Analysis of the serum levels of insulin (ng/ml) (**c**) and glucose (mg/dL) (**d**) in *db/m* and *db/db* mice at 9, 11, and 19 weeks of age ($n$ = 5 mice each). **e** Representative immunoblotting showing Orai1 and TRPC6 expression in the glomerulus of *db/m* and *db/db* mice at 11 and 19 weeks. **f** Densitometry of Orai1 (left) and TRPC6 (right) in **e**. Relative (Rel.) protein levels were normalized to *db/m* mice (11 weeks of age) $n$ = 3 independent experiments. **g** Representative immunofluorescence images of synaptopodin (synap.; green) and Orai1 (red) in the glomeruli of *db/m* and *db/db* mice. Immunostaining was repeated five times independently from 5 mice each. Scale bar = 25 μm. **h** Immunoblotting analysis showing synaptopodin expression in the glomeruli of *db/m* and *db/db* mice at 11 and 19 weeks. **i** Quantification of the western blots in **h**. $n$ = 3 independent experiments. **j** Effect of cyclosporin A (CsA, *i.p.* injection) on urine Alb/Cr ratio (left) and 24 h urinary albumin excretion (right) in *db/m* and *db/db* mice ($n$ = 5–7 images from mice each). **k** Immunofluorescence staining showing the effect of CsA on synaptopodin dissolution in *db/db* mice. Scale bar = 25 μm. **l** Summary of synaptopodin quantification in **k**. $n$ = 30 images from 5 to 10 mice each. **m** Representative TEM images of foot processes from *db/m* and *db/db* mice with/without administration of CsA. Magnification is ×30,000. Scale bar = 1 μm. **n, o** GBM thickness (**n**) and an average number of foot processes per 1 μm length of GBM (**o**) in *db/m* and *db/db* mice with/without administration of CsA. Thirty images from 5–7 mice in each group. Bar graph data are expressed as mean ± SEM and were analyzed using two-way ANOVA followed by Tukey's multiple comparisons test. **$p < 0.01$, ****$p < 0.0001$; ns not significant.

term insulin action targeting Orai1 on the actin rearrangement of podocytes. Actin reorganization by the acute stimulation of insulin (~30 min) was restored after recovery, whereas long-term exposure of insulin (>6 h) led to irreversible cytoskeletal changes (Fig. 5a, b). The chronically elevated insulin levels may reflect the situation in obese insulin-resistant individuals showing proteinuric glomerular lesions. To explore the long-term consequence of insulin stimulation, we used the hyperinsulinemic *db/db* mouse model, which exhibits increased circulating insulin levels at the early stage and then progresses to type 2 diabetes[15]. We confirmed that these mice showed high circulating levels of insulin at the early stage (11 weeks of life) that decreased at the late advanced stage (19 weeks), possibly due to islet β-cell failure (Fig. 5c, d). Orai1 overexpression was prominent only at the early

stage coinciding with hyperinsulinemia, while a higher expression of TRPC6 was obvious at the late stage (Fig. 5e, f). Immunofluorescence images showed higher Orai1 expression in glomerular cells obtained from 11-weeks-old *db/db* mice than those obtained from *db/m* control mice (Fig. 5g). As a downstream effector of the $Ca^{2+}$/calcineurin pathway, synaptopodin levels remained decreased at both 11 and 19 weeks of age in the *db/db* mice, suggesting that $Ca^{2+}$-triggered podocyte injury is extended after Orai1 decline and is possibly related to the action of TRPC6 (Fig. 5h, i).

**Inhibition of $Ca^{2+}$-calcineurin pathway ameliorates foot process disruption and albuminuria in *db/db* mice.** Based on our

data, insulin-activated SOCE can accelerate $[Ca^{2+}]_i$ overload, which in turn causes actin remodeling, disruption of the slit diaphragm, and glomerular basement membrane (GBM) adhesion leading to proteinuria[3]. A previous report suggests that intensive insulin therapy can induce albuminuria[36], which raises the question of whether hyperinsulinemia per se contributes to the disruption of glomerular filtration mediated by SOCE-derived $[Ca^{2+}]_i$ overload. We examined whether the activation of the $Ca^{2+}$/calcineurin pathway worsens podocyte function in hyperinsulinemic db/db mice. Administration of CsA as a calcineurin inhibitor markedly attenuated albuminuria in db/db mice (Fig. 5j). Consistent with acute insulin effects in cultured podocytes and transgenic Orai1-overexpressing mice, synaptopodin dissolution in db/db mice was also rescued by CsA (Fig. 5k, l). The main structural features observed in the glomeruli of db/db mice were foot process effacement and altered GBM thickness. All these pathological alterations were also restored by CsA (Fig. 5m–o and Supplementary Fig. 8a), demonstrating that the inhibition of the $Ca^{2+}$/calcineurin pathway ameliorates slit-diaphragm disruption and albuminuria in chronic hyperinsulinemic db/db mice.

**SOCE inhibition protects podocyte slit diaphragm and albuminuria in db/db mice.** Given that our results showed that podocyte-specific Orai1 ablation in vivo prevents insulin-driven albuminuria and that the inhibition of calcineurin protects podocyte slit diaphragm and albuminuria in hyperinsulinemic db/db mice, we further confirmed the critical role of Orai1 in $Ca^{2+}$-calcineurin-dependent podocyte dysfunction and albuminuria in db/db mice using pharmacological inhibitors. Consistent with Orai1 overexpression, SOCE was also upregulated in the isolated glomeruli of db/db mice compared to those of the control db/m mice (Fig. 6a, b). Blockade of Orai1 by GSK-7975A or 2-APB administration markedly ameliorated albuminuria in db/db mice (Fig. 6c). Moreover, synaptopodin dissolution was also relieved by the pharmacological inhibition of Orai1 in db/db mice (Fig. 6d–g) supporting the assumption that Orai1-activated $Ca^{2+}$-calcineurin signaling could be responsible for synaptopodin degradation in the glomeruli of hyperinsulinemic mice. Furthermore, foot process effacement and altered GBM thickness in db/db mice were also rescued by GSK-7975A and 2-APB administration (Fig. 6h–j and Supplementary Fig. 8b). Taken together, our results demonstrate that the blockade of the Orai1-calcineurin pathway can ameliorate slit-diaphragm disruption and albuminuria in hyperinsulinemic db/db mice (Fig. 7).

## Discussion

Growth factors alter $[Ca^{2+}]_i$ regulation via RTK signaling that can affect podocyte actin remodeling and slit-diaphragm dysfunction leading to proteinuria[3,10]. Our study demonstrates that Orai1 regulation in podocytes is a key $Ca^{2+}$ influx mechanism regulated by growth factors and responsible for hyperinsulinemia-induced albuminuria. However, insulin and other growth factors are critical for podocyte survival and the suppression of insulin signaling is known to impair the glomerular filtration barrier[11–13,17–19]. Therefore, well-balanced insulin RTK signaling is required for maintaining podocyte actin cytoskeleton and glomerular filter integrity to prevent proteinuria.

Several observations argue whether Orai1 overexpression protects or aggravates against renal pathology[37–40]. Orai1 activation in mesangial cells protects against matrix protein expression under diabetic conditions[37,40]. In contrast, the overexpression of Orai1 in the epithelial cells of the proximal tubule aggravates fibrosis[38] and renal cell carcinoma progression[39]. Orai1 promotes lymphocyte IL17 expression in $CD4^+$ cells and, in turn, contributes to

acute kidney injury (AKI) and AKI-to-CKD progression[41]. All these results imply that the pathophysiological role of Orai1 and SOCE is cell type-specific in a context-dependent manner in renal pathologies. Our study clearly demonstrates that Orai1 upregulation by transgene overexpression and/or insulin stimulation aggravates perturbation of the glomerular slit diaphragm leading to albumin leakage. Additionally, our results support that growth factor signaling targeting Orai1 is critical for $[Ca^{2+}]_i$ homeostasis and the selective filtration function of podocytes under various pathophysiological conditions.

Multiple studies have provided compelling evidence that $[Ca^{2+}]_i$ links actin dynamics to podocyte dysfunction and proteinuria[3,4]. Among $Ca^{2+}$-permeable ion channels, TRPC5 and TRPC6 channels play a crucial role in $[Ca^{2+}]_i$ regulation and the actin dynamics of podocytes[6–9,25,26,42]. TRPC5 and TRPC6 are antagonistic regulators of actin dynamics and cell motility in podocytes[3,26]. Under physiological conditions, angiotensin II receptor stimulation activates both TRPC5 and TRPC6 with different phenotypes associated with actin dynamics and cellular behaviors. TRPC5-mediated $Ca^{2+}$ influx decreases stress fiber assembly and promotes cell migration, whereas TRPC6 activation promotes stress fiber formation and a contractile cell phenotype. In the present study, however, Orai1-mediated $Ca^{2+}$ influx decreased stress fiber formation and increased motility and contractility, demonstrating the distinct role of Orai1 in actin dynamics compared to that of TRPC5 and TRPC6 in podocytes. Hence, dissecting the $Ca^{2+}$ microdomain and its downstream effectors by Orai1 and TRPC5/6 channels for linking the modulation of actin dynamics to podocytopathy requires further investigation.

While multiple studies have reported SOCE upregulation mediated by growth factors[20–23], its underlying molecular mechanism has not been identified. Here, we uncover that insulin activates the surface trafficking of Orai1 via PI3K-stimulated VAMP2-dependent vesicular exocytosis in mouse podocytes. Insulin increases the cell-surface abundance of TRPC6, but not TRPC5, via a NADPH oxidase-dependent mechanism[27]. We had proposed in previous reports that IGF-1 and serum stimulate cell-surface abundance of both TRPC6 and Orai1 via a PI3K-dependent pathway[8,32,35]. However, in this study, insulin upregulated Orai1-dependent SOCE and albuminuria, which were unaffected by the genetic suppression of TRPC6 (Supplementary Figs. 1 and 4). Our data, along with the previous findings, suggest that growth factors can modulate $Ca^{2+}$ influx through both TRPC6 and Orai1 independently; however, Orai1 might be mainly responsible for insulin-stimulated SOCE activation and cytoskeletal rearrangement in podocytes based on evidence from the in vitro and in vivo experiments.

Recently, Zeng et al. showed higher expression of Orai isoforms in proximal tubules, which were downregulated in type I diabetic nephropathy (DN). Moreover, inhibition of Orai1 aggravated proteinuric pathologies in type I DN[43]. Different from this report, we observed that Orai1 is upregulated in the glomeruli of a type 2 DN animal model, that is, the db/db mice. The db/db mice display high levels of circulating insulin without showing a serious diabetic phenotype up to 8 weeks of age, and blood glucose levels gradually increase, under the action of high but inadequate insulin levels, till 14 weeks[15]. Eventually, these mice develop advanced diabetes with a deteriorated β-cell insulin secretion by the 20th week. Consistent with this description, we observed that db/db mice showed high plasma insulin levels along with increased Orai1 expression, synaptopodin degradation, and albuminuria at 11 weeks of age, supporting that Orai1 upregulation could be an early event of proteinuric filter dysfunction in the hyperinsulinemic stage of type 2 DN animal model. Together, our data and previous findings demonstrate that the tightly

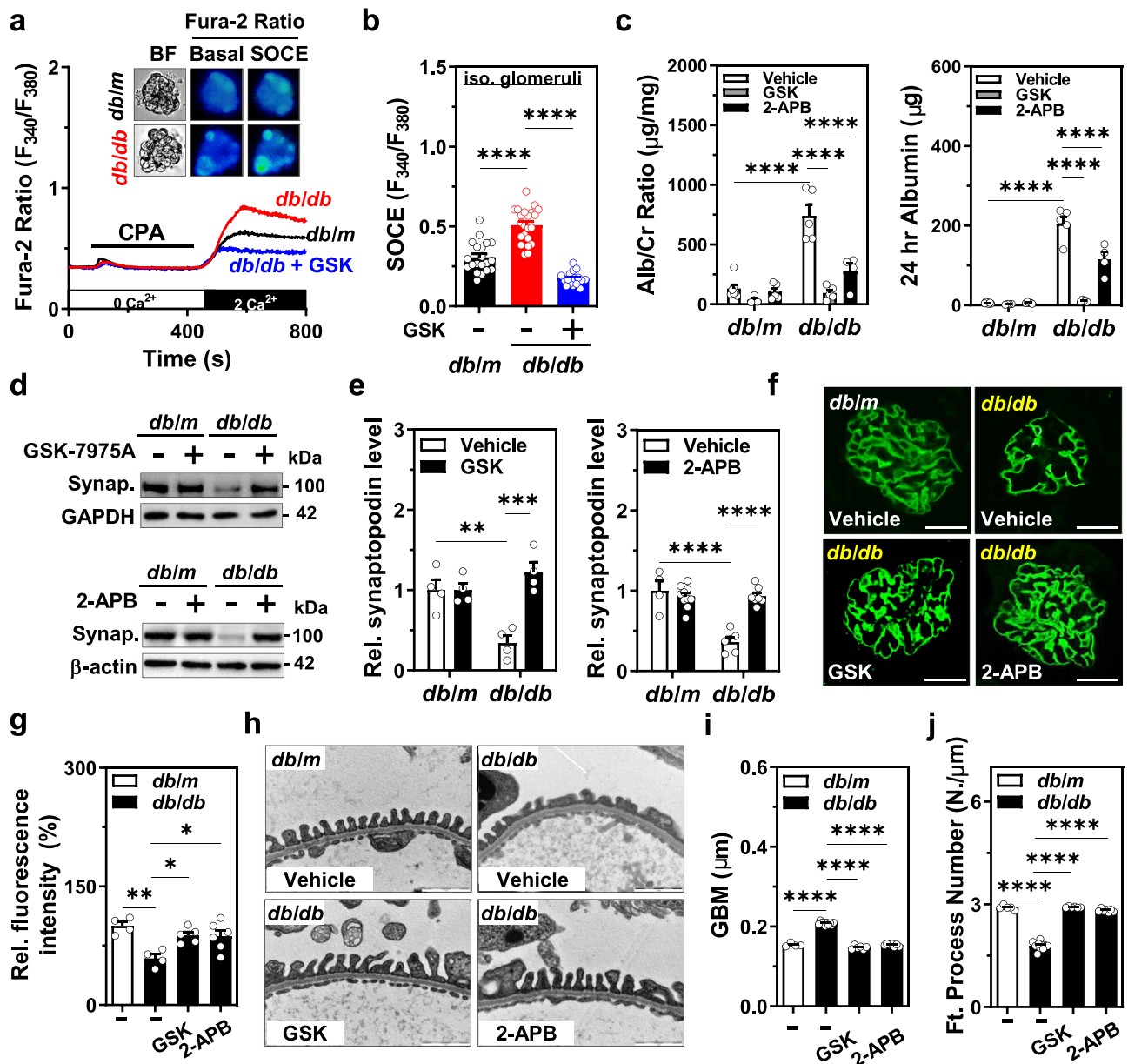

**Fig. 6 Orai1 inhibition protects podocyte foot processes and proteinuria. a** Representative traces of fura-2 ratio and the images of bright-field and fura-2 fluorescence microscopy (upper panel) of podocytes, carried out in situ on the acutely isolated mouse in *db/m* and *db/db* mice treated with GSK-7975A (5 mg/kg/day for 1 month). **b** Summary of the SOCE in **a**. $n = 20$ (*db/m* + Vehicle), 22(*db/db* + Vehicle), and 17 (*db/db* + GSK) glomeruli per each group, respectively. **c** Effect of SOCE inhibitors on albuminuria. Twenty-four-hour urinary Alb/Cr ratio (left) and albumin excretion (right) were measured in *db/m* and *db/db* mice. Vehicle (PBS), GSK-7975A (GSK, 5 mg/kg/every 2 days) or 2-APB (2 mg/kg/every 2 days) were inoculated *i.p.* from 8 to 12 weeks. $n = 7$ (*db/m* + Veh), 4 (*db/m* + GSK), 5 (*db/m* + 2-APB), 5 (*db/db* + Veh), 5 (*db/db* + GSK), and 4 (*db/db* + 2-APB). **d** Representative immunoblots representing the effects of SOCE inhibitors, GSK-7975A (upper) or 2-APB (lower), on synaptopodin expression from *db/m* and *db/db* mice. **e** Summary of quantification in **d**. $n = 4$ mice each (left). $n = 4$ (*db/m* + Veh), 10 (*db/m* + 2-APB), 5 (*db/db* + Veh), and 7 (*db/db* + 2-APB) (right). **f** Rescue of synaptopodin depletion by SOCE inhibitors (GSK or 2-APB). Synaptopodin staining (green) of mouse glomeruli from *db/m* and *db/db* mice with or without GSK or 2-APB *i.p.* injection. Scale bar = 25 μm. **g** Analysis of the relative fluorescence intensity in **f**. $n = 4$ (*db/db* + Veh), 4 (*db/db* + Veh), 5 (*db/db* + GSK), and 6 (*db/db* + 2-APB). Thirty images from mice in each group. **h** Representative TEM images of the foot processes of the podocytes of SOCE inhibitor-treated *db/db* mice. Magnification is ×30,000. Scale bar = 1 μm. **i, j** Summary of GBM thickness (**i**) and an average number of foot processes per 1 μm length of GBM (**j**) in **h**. $n = 4$ (*db/m* + Veh), 10 (*db/db* + Veh), 6 (*db/db* + GSK), and 7 (*db/db* + 2-APB) thirty images from mice in each group. Data are expressed as mean ± SEM and were analyzed using two-way ANOVA followed by Tukey's multiple comparisons test. $*p < 0.05$, $**p < 0.01$, $****p < 0.0001$; ns not significant.

balanced insulin signaling targeting Orai1-mediated SOCE is critical for inducing proteinuria in both type I and type II DN.

$[Ca^{2+}]_i$ signaling can, in turn, activate the calcineurin regulating podocyte actin-binding protein, synaptopodin, whose degradation is related to proteinuric diseases[28–30]. Accumulating

evidence indicates that calcineurin inhibitors ameliorate proteinuria by inhibiting synaptopodin degradation[28–30,42]. Our data demonstrate that Orai1 activation by insulin induces actin remodeling and albuminuria via the $Ca^{2+}$/calcineurin/synaptopodin pathway. Actin remodeling by short-term insulin

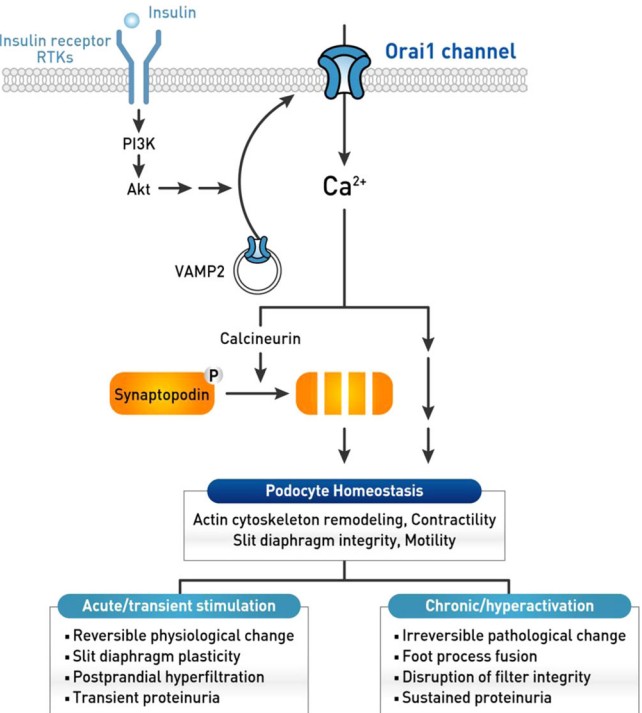

**Fig. 7 Working model of the effect of insulin signaling on podocyte homeostasis by the targeting of podocyte Orai1.** Insulin stimulates Orai1 leading to an aberrant $[Ca^{2+}]_i$ signaling, which in turn activates calcineurin and causes synaptopodin degradation, actin cytoskeleton remodeling, increased contractility, and motility, and dysfunction of slit-diaphragm integrity. Acute (or short-term) stimulation of Orai1 by insulin receptor or receptor tyrosine kinases causes reversible physiological changes in the podocyte actin cytoskeleton. This may contribute to slit-diaphragm plasticity and/or survival against stress or injury. In contrast, chronic stimulation or hyperactivation of Orai1 activation leads to irreversible pathological changes such as podocyte foot process effacement and severe proteinuria.

stimulation is reversible, raising the possibility that insulin signaling may contribute to podocyte slit-diaphragm plasticity and survival. In contrast, chronic hyperinsulinemic conditions may cause irreversible actin cytoskeletal changes associated with synaptopodin degradation triggered by sustained $Ca^{2+}$ entry and calcineurin activation in podocytes. The present study shows that severe foot process effacement and albuminuria in hyper-insulinemic *db/db* mice are ameliorated by inhibition of Orai1 or calcineurin, which could provide therapeutic targets to preserve the glomerular filtration barrier (Fig. 7).

In summary, we show that increased expression and/or activity of podocyte-specific Orai1 by insulin perturbs the glomerular filter suggesting that fine-tuning of podocyte $[Ca^{2+}]_i$ via Orai1 plays a pivotal role for podocyte homeostasis and filter function. The molecular mechanisms causing podocyte foot process effacement and proteinuria remain unclear, and there is a lack of targeted therapies for the proteinuric renal disease. The data presented in this study shed important insights on the patho-physiological role of $[Ca^{2+}]_i$ perturbation by insulin RTK signaling, linking actin remodeling to podocyte foot process effacement and proteinuria. Taken together, our results provide a novel perspective on growth factor signaling by demonstrating the involvement of Orai1 in the pathogenesis of podocyte dysfunction and suggest new possible therapeutic strategies for proteinuric kidney diseases such as diabetic nephropathy.

## Methods

**Reagents**. Unless otherwise noted, all chemicals and reagents were purchased from Sigma-Aldrich (St Louis, MO, USA). GSK-7975A and LY294002 were purchased from AOBIOUS Inc. (Gloucester, MA, USA) and Calbiochem (San Diego, CA, USA), respectively. Cyclosporin A (Cipol-N®) was obtained from Chong Kun Dang Pharmaceutical Corporation (Seoul, Korea).

**Cell culture, transfection, and knockdown by small interfering RNA**. The immortalized mouse podocyte cell line (a kind gift from Dr. Peter Mundel, Harvard Medical School, USA) and HEK293FT (human embryonic kidney 293) cells were cultured as previously described[8]. All cells tested as "mycoplasma-negative." All DNA plasmids were transfected using X-tremeGENE® HP DNA transfection reagent (Roche, Mannheim, Germany) according to the manufacturers' instructions. Experiments were conducted 48 h after transfection. Mouse siRNAs for Orai1 (M-056431-01), Orai2 (M-057985-01), Orai3 (M-054417-01), STIM1 (M-062376-01), STIM2 (M-055069-01), and VAMP2 (M-041975-01) were obtained from Dharmacon (Chicago, IL, USA). Non-targeting control oligonucleotide (sc37007) and mouse siRNA for TRPC6 (sc42673) were provided by Santa Cruz Biotechnology (Santa Cruz, CA, USA). Transfection of siRNA oligonucleotide was performed using DharmaFECT siRNA transfection reagent (Thermo Scientific, Lafayette, CO, USA) according to the manufacturer's instructions.

**Animal studies**. To generate conditional *Orai1* knockout (KO) mice, mice homozygous for the Orai1 allele flanked by the loxP site (*Orai1^{fl/fl}*) were back-crossed to C57/BL6/J mice for at least ten generations. *Orai1f^{l/-}* mice and Podocin-cre (*Nphs2.Cre+ = B6.Cg-Tg(NPHS2-cre)295Lbh/J*) mice were generated as described previously[44,45]. *Trpc6^{-/-}* mice have been described previously[32]. Adult (9–11-weeks old) male KO mice of *Orai1* and *Trpc6* from the same litter were grouped in all experiments.

Six-week-old male BKS.Cg-m+/+Lepr^{db}/BomTac *db/m* and *db/db* mice were purchased from Taconic Farms (Germantown, NY, USA); *db/m* mice were used as controls in all experiments. All mice were checked for fasting blood glucose levels, body weight, and food intake once every week during maintenance. Mice were inoculated daily with vehicle (PBS), cyclosporin A (CsA) (10 mg/kg, every 2 days; from weeks 9 to 15), 2-APB (2 mg/kg, every 2 days; from weeks 8 to 14), and GSK-7975A (5 mg/kg, every day; from weeks 8 to 14). All mice were raised in the individual ventilated cage (IVC) racks at a constant temperature (22 ± 3 °C) and constant relative humidity (50 ± 10%) using fluorescent lamps (lights are on 6:00 to 18:00) for 12 h and fed with solid feed 5L79® (LabDiet, St. Louis, MO, USA). All mice were maintained in pathogen-free barrier facilities. Animal experiments were performed according to the eighth edition of the Guide for the Care and Use of Laboratory Animals (NRC 2011) and all experimental protocols involving mice were approved by the Yonsei University Wonju College of Medicine Institutional Animal Care and Use Committee (YWC-170919-1, YWC-130826-2, and YWC-161222-1).

**Isolation of glomeruli**. Glomeruli were isolated from mice as previously described[46] with some modifications. Briefly, adult male mice were anesthetized with ketamine and rompun (100 and 10 mg/kg, respectively) and perfused with cold phosphate-buffered saline (PBS). The renal cortex was cut into small fragments ranging from 1 to 2 mm in size, dissociated using a sieve-tissue grinder (CD1-1KT, Sigma-Aldrich) and pushed through the 200-μm (s4145, Sigma-Aldrich) and 40-μm meshes (Greiner Bio-One, Vilvoorde, Belgium) to obtain rat and mice glomeruli, respectively. The glomerular fractions were collected and seeded on poly-L-lysine coated coverslips in a $Ca^{2+}$-free physiological salt solution containing (in mM) 135 NaCl, 5 KCl, 1 MgCl$_2$, 1 EGTA, 10 HEPES, and 5.5 glucose (pH 7.4). After visual inspection, the glomeruli were cultured in RPMI 1640 medium supplemented with 1% penicillin. After 6 h, the media was replaced with low glucose Dulbecco's modified Eagle's medium (DMEM) supplemented with 10% fetal bovine serum (FBS, 16000-044, GIBCO, Grand Island, NY, USA) and 1% penicillin. Fura-2 $Ca^{2+}$ imaging of isolated glomeruli was performed after 24 h of seeding.

**Immunofluorescence**. Immunofluorescence staining of mouse kidney tissues was performed as previously described[8] with a few modifications. The tissues were fixed with 4% paraformaldehyde (PFA) overnight at 4 °C, stored in 30% sucrose, and embedded with the OCT compound for preparing slides by cryosection. Following permeation and blocking steps, the slides were incubated with goat anti-synaptopodin (1:40 dilution; sc21537, Santa Cruz Biotechnology), rabbit anti-Orai1 (1:100 dilution, NBP1-46470, Novus, Littleton, CO, USA), and rabbit anti-Flag-tag (1:20 dilution; A00170, GenScript) at 4 °C overnight and subsequently incubated with anti-goat-Alexa 488 (1:200 dilution; A-11055, Invitrogen, Carlsbad, CA, USA), anti-goat-Cy3 (1:200 dilution; 705-165-003, Jackson ImmunoResearch), and anti-rabbit-Alexa 594 (1:200 dilution; A-21207 Invitrogen, Carlsbad, CA, USA) for 1 h at room temperature (approximately 22–25 °C).

For the immunofluorescence staining of cultured mouse podocytes or isolated glomeruli, cells were grown on type-1 collagen (Invitrogen, Carlsbad, CA, USA)-coated coverslips and fixed with 4% paraformaldehyde in PBS for 15 min at 37 °C,

and the isolated glomeruli were fixed with chilled 100% acetone for 10 min at −20 °C. Phalloidin staining (1:40 dilution; A12381 Invitrogen, CA, USA) was performed as previously described[8]. Specific antibodies against rabbit anti-Orai1 (1:100 dilution, ab86748, Abcam, Cambridge, MA, US), goat anti-nephrin (1:50, AF3159, R&D systems), goat anti-synaptopodin (1:40 dilution, sc21537), or rabbit anti-paxillin (1:250 dilution, ab32084, Abcam, Cambridge, MA, US) were incubated overnight at 4 °C followed by incubation with a secondary antibody, anti-rabbit-Alexa 594 (1:200 dilution; A-21207 Invitrogen, Carlsbad, CA, USA), anti-goat-Alexa 488 (1:200 dilution; A-11055, Invitrogen, Carlsbad, CA, USA), or anti-rabbit-Alexa 488 (1:200 dilution; 711-545-152, Jackson ImmunoResearch) for 1 h. Images were obtained using a laser scanning confocal microscope (TCS SPE, Leica Microsystems GmbH, Wetzlar, Germany; or LSM 800, Zeiss, Jena, Germany). The fluorescence intensities of the images were quantified using ZEN 2.3 software (Zeiss, Jena, Germany) or Image J 1.53c (NIH, Bethesda, Maryland, USA).

**Histochemistry and transmission electron microscopy (TEM)**. Histochemistry was performed as previously described[8,39] with some modifications. Four-micrometer sections of the paraffin blocks of mouse kidney tissues were cut and attached on coated slides using an automatic immunostaining machine (Ventana Benchmark XT, Roche Diagnostics, Basel, Switzerland). The sections were stained with hematoxylin and eosin (Roche Diagnostics). TEM was performed as previously described[8]. The TEM images were analyzed by a renal pathologist in a blinded manner and results were analyzed using one- or two-way ANOVA followed by Dunnett's or Tukey's multiple comparison tests.

**Quantitative real-time polymerase chain reaction (qPCR)**. Total RNA was purified from the trypsinized pellets of the podocyte cell line and the isolated glomeruli from the kidneys of $db/m$ and $db/db$ mice using the Hybrid-R™ total RNA purification kit (305-101, GeneAll, Seoul, Korea). Complementary DNA was synthesized from 1 μg of the total RNA using the ReverTraAce® qPCR RT Master Mix with gDNA Remover (FSQ-301, Toyobo, Osaka, Japan) according to the manufacturer's instructions. Real-time PCR was performed to measure mRNA expression for the genes of interest using sequence-specific mouse primers (Supplementary Table 1) with a specific DNA-intercalating fluorescence dye, Power SYBR® Green PCR Master Mix (4367659, Applied Biosystems, Waltham, MA USA). 18S was used as the internal control. To analyze the expression of each gene, experiments were conducted in triplicate in a real-time PCR system (7900HT, Applied Biosystems, Foster City, CA, USA). Data were analyzed following the $2^{-\Delta\Delta CT}$ method with 18S as the reference gene.

**Measurement of intracellular Ca$^{2+}$ concentration**. Intracellular Ca$^{2+}$ ($[Ca^{2+}]_i$) measurement was performed as described previously[8,39]. The normal physiological salt solution (PSS) used for bath perfusion contained (in mM) 135 NaCl, 5 KCl, 1 MgCl$_2$, 2 CaCl$_2$, 10 HEPES, and 5.5 glucose (pH 7.4). The Ca$^{2+}$-free PSS included (in mM) 135 NaCl, 5 KCl, 1 MgCl$_2$, 1 EGTA, 10 HEPES, and 5.5 glucose (pH 7.4). Cyclopiazonic acid (c1530, Sigma-Aldrich) was dissolved in Ca$^{2+}$-free PSS for ER Ca$^{2+}$ release. Data were analyzed using the MetaFluor 6.1 software (Sutter Instrument, Novato, CA, USA). All $[Ca^{2+}]_i$ measurements were performed at 37 °C using a heat controller (Warner Instruments, Hamden, CT, USA).

**Patch-clamp recording**. Cultured podocytes show a flattened shape with a relatively high membrane capacitance indicative of large cell size (range 90–200 pF for podocytes vs ~12 pF for HEK293FT cells). This feature can create space-clamp errors and overcompensation problems. To reduce the space-clamp errors associated with podocytes, we measured CRAC currents in cultured podocytes within 2 h after attachment to elicit spherical and relatively smaller cells. For the accurate analysis of the intrinsic channel properties of CRAC currents regulated by insulin, we used HEK293FT cells expressing mCherry-Flag-tagged Orai1 and YFP-STIM1 (0.2 μg each per 35-mm dish) that are much smaller in size (mean membrane capacitance: approximately 12 pF). Currents were recorded using the whole-cell-dialyzed configuration of the patch-clamp technique as described previously[8]. Whole-cell currents were recorded under voltage-clamp using an EPC-9 patch-clamp amplifier (Heka Electronik, Lambrecht, Germany). The patch electrodes were coated with silicone elastomer (Sylgard 184; Dow Corning, Midland, MI, USA) and fire-polished and showed resistances of 2–3 MΩ when filled with the pipette solution. The cell membrane capacitance and series resistance were compensated (>80%) electronically using an EPC9 amplifier. Data acquisition was performed using the Pulse/Pulsefit (v8.50) software (Heka Electronik). All electrophysiological recordings were performed at room temperature (approximately 22–25 °C). The bath and pipette solutions contained (in mM) 130 NaCl, 5 KCl, 10 CaCl$_2$, 2 MgCl$_2$, 10 HEPES, and 10 glucose (pH 7.4) and 140 Cs-Asp, 4 BAPTA, 6 EGTA, 6 MgCl$_2$, and 10 HEPES (pH 7.2), respectively.

**Western blotting and cell-surface biotinylation assay**. We conducted western blot analysis and cell-surface biotinylation assay as described previously[8,39]. Primary antibodies for VAMP2 (1:1000 dilution, ab181869), Orai1 (1:1000 dilution, ab86748), and β-actin (1:5000 dilution, ab6276) were obtained from Abcam. STIM1 (1:1000 dilution, 610954) and Flag-HRP (1:5000 dilution, A8592) were purchased from BD Biosciences (Clontech, Palo Alto, CA, USA) and Sigma-

Aldrich, respectively. Antibodies against Akt (1:2000 dilution, 9272), p-Akt$^{Ser473}$ (1:2000 dilution, 9271), and p-Akt$^{Thr308}$ (1:2000 dilution, 2965) were obtained from Cell Signaling Technology (Beverly, MA, USA). GAPDH (1:10,000 dilution, sc25778), and antibodies against synaptopodin (1:1000 dilution, sc21537) were purchased from Santa Cruz Biotechnology. Antibodies against Orai2 (1:1000 dilution, ACC-061), Orai3 (1:1000 dilution, ACC-065), STIM2 (1:100 dilution, ACC-064), and TRPC6 (1:500 dilution, ACC-120) were obtained from Alomone Labs (Jerusalem, Israel).

**In vitro scratch assay**. Podocytes were plated at a density of $1 \times 10^6$ cells per well in type-1 collagen-coated 6-well plate and grown to 100% confluence. Cells were scratched with a 1-μl pipette tip across the center of the wells. To distinguish cell migration from proliferation, all scratch assays were performed in the presence of anti-tumor drug mitomycin C (0.5 μg/ml, M4287, Sigma-Aldrich). The images were captured using a microscope 24 h after drug treatment (time 0, initial time point).

**Collagen contraction assay**. Collagen-based Contraction Assay Kit was purchased from Cell Biolabs (CBA-201, San Diego, CA, USA) and used according to the manufacturer's instructions. Podocytes were harvested and the pellet was suspended in a serum-free low glucose DMEM at a density of $5 \times 10^6$ cells/ml. For each assay, 100–150 μl of the cell suspension was mixed with 400 μl of neutralized collagen solution and added to one well of a 24-well cell culture plate and allowed to polymerize for 1 h at 37 °C. After polymerization, 1 ml of complete media was added to the top of each collagen gel. The gels were released by running a sterile pipette tip along the sides of the well after 48 h. The culture dish was scanned immediately after being released (time 0, initial time point) and at 24 h.

**In vitro albumin permeability assay**. In vitro albumin permeability assay was performed as described previously[8]. To detect macromolecular passage across the membrane, cell permeability was determined by measuring the transepithelial passage of FITC-labeled BSA (A9771, Sigma-Aldrich) from the apical to the basolateral compartment of the trans-well bicameral chambers (0.4 μm pore; 3516, Corning Costar Corporation, Cambridge, MA, USA). Confluent differentiated podocytes were exposed to insulin (100 nM) for 12 or 24 h. In the upper compartment, the culture medium was replaced with 2 ml of FITC-BSA (250 μg/ml), while the lower volume was refreshed with 2 ml of complete medium. At different diffusion times (0.1, 0.5, 1, 1.5, 2, and 12 h), 100 μl samples were drawn from the lower compartment, which was replenished with an equal volume of fresh medium each time. The collected samples were measured using fluorescence spectroscopy at 450 nm wavelength. The albumin concentrations in the samples were calculated using linear regression of a diluted series of the tracer.

**Urinary albumin and creatinine measurement**. For the determination of urinary albumin excretion, mice were placed into individual metabolic cages for 24 h and urinary albumin was measured using an enzyme-linked immunoassay specific for mouse albumin (Albuwell M; Exocell, Philadelphia, PA, USA). To avoid errors arising from incomplete urine collection, albumin excretion was normalized to urine creatinine levels (Exocell).

**Blood glucose and insulin measurement**. The blood glucose concentration was estimated by the glucose oxidase method using a commercial glucometer and test strips (Accu-chek Active™ Test meter). Blood samples were obtained from the retro-orbital vein plexus of the animal following 18 h of starvation. Approximately, 100–150 μl of blood samples were obtained from each mouse. The plasma insulin levels were determined using the Morinaga Ultra-Sensitive Mouse/Rat Insulin ELISA Kit (Morinaga Institute of Biological Science, Inc, Kanazawa, Japan).

**In vivo gene delivery of Orai1**. Hydrodynamic gene delivery (MIR 5340, Mirus Bio LLC, Madison, WI, USA) was achieved by tail vein injection as previously described[8], with some modifications. Plasmid DNA encoding mCherry-Flag-tagged Orai1 vector (10 μg) dissolved in 2 ml isotonic saline was injected via the tail vein into 5-weeks-old mice (BKS.Cg-m+/+Leprdb/BomTac). Control animals received plasmid DNA encoding mCherry in the same volume. The mice were then inoculated i.p. with vehicle (DMSO) or insulin (5 U/kg) at 24 h post-gene delivery and 24 h after urine was collected in a metabolic cage. Next, mice were euthanized and the efficiency of the gene transfer was monitored in isolated glomerular tissue using western blot analysis and double-labeling immunofluorescence microscopy of Flag and synaptopodin. All animal protocols were approved by the Yonsei University Wonju College of Medicine Institutional Animal Care and Use Committee (YWC-130826-2).

**Statistics and reproducibility**. Data analysis was performed using the GraphPad Prism software (version 9, GraphPad Software, San Diego, CA). Statistical comparisons between two groups of data were performed using a two-tailed unpaired Student's $t$-test. Multiple comparisons were conducted using one- or two-way ANOVA followed by Dunnett's or Tukey's multiple comparisons test. $P$-values < 0.05 and 0.01 were considered statistically significant. Normal distribution of data

was assumed, and data are presented as mean ± SEM. Further statistical details are provided in figure legends and in the Source Data file. All experiments have been reproduced in at least three independent experiments, unless otherwise specified in the figure legends.

**Reporting summary**. Further information on research design is available in the Nature Research Reporting Summary linked to this article.

## Data availability

The authors declare that all data supporting the findings of this study are available within the paper, Supplementary Information files, and the Source Data files. Source Data files including original data of $Ca^{2+}$ imaging, animal experiment, and western blots have been deposited in Figshare.com (https://doi.org/10.6084/m9.figshare.16618213). All other data supporting the findings of this study are available from the corresponding author on a reasonable request. Source data are provided with this paper.

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

## Acknowledgements

We thank professors Shmuel Muallem (NIH), Joseph Yuan (University of North Texas), Joo Young Kim (Yonsei University), Chan Young Park (UNIST), and Peter Mundel (Harvard Medical School) for materials. We are grateful to professors Claes B Wollheim, Nicolas Demaurex (University of Geneva), Andreas Wiederkehr (Nestle Institute of Health Science), and Chou-Long Huang (University of Iowa) for critical reading and their comments on the manuscript. This study was supported by the Medical Research Center Program (2017R1A5A2015369) and the Basic Science Research Program (NRF-2010-0024789, 2013R1A1A2060764, 2015R1D1A1A01060454, 2017R1D1A3B03031760, and 2019R1A2C1084880) through the National Research Foundation of Korea, and the Intramural Research Program of the NIH (Z01-ES-101684 to L.B.).

## Author contributions

J.-H.K. designed the study, conducted the experiments, analyzed the data, and participated in writing the paper; K.-H.H., and B.T.N.D. conducted the experiments and analyzed data; M.E. conducted and analyzed the pathology experiments including IHC and TEM; I.D.K designed and analyzed the electrophysiological data; Y.G., S.Y., H.Y.G., and L.B. generated the genetically engineered mice and analyzed the animal data; K.-S.P., and S.-K.C. designed and supervised the entire project and wrote the final manuscript. All authors have read and approved the paper.

## Competing interests

The authors declare no competing interests.
