## [Peer Review File · Nature Communications]

Insulin-activated store-operated Ca^{2+} entry via Orai1 induces podocyte actin remodeling and causes proteinuriaReviewers' comments:

Reviewer #1 (Remarks to the Author):

The authors present an interesting paper on the function of podocytes which perform essential roles in glomerular filtration. There have been many reports on growth factor and calcium signaling controlling glomerular filter integrity but considerable uncertainty still exists about the calcium channels involved. The paper addresses how insulin signaling might affect store-operated calcium entry in podocytes and hence regulate actin cytoskeleton dynamics and glomerular filtration. In the current work, the authors postulate that the store-operated calcium channel, Orai1, is an important molecular component of SOCE in podocytes and may affect and control proteinuria. They show that Orai1 is activated by insulin signaling which alters actin cytoskeleton dynamics leading to increased transepithelial albumin leakage via the calcium-calcineurin pathway. They look at the effects of transgenic overexpression of Orai1 in mice which they show induces foot process fusion and albuminuria and this can be decreased by inhibition of the Orai1-calcineurin pathway. The authors discuss the implications of the possible actions of Orai1 and suggest that tightly balanced insulin signaling by podocyte Orai1 is important for maintaining filter integrity. The work does offer interesting new perspectives on the pathogenesis of podocyte injury and possible therapeutic strategies for counteracting proteinuric diseases.

While the paper is interesting, perhaps the biggest question is whether the results on Orai1 in podocytes are novel and important as opposed to just making some new correlative inferences on the role of well-established signaling pathways, but this time in podocytes. Certainly, much in the paper is not exactly novel. For example, there are a number of papers in the literature linking insulin signaling through mTORC2 and Akt to changes in actin polymerization. There are also many studies linking mTORC2 and Akt to exocytosis, for example of the GLUT4 transporter by AS160. The novel part of the paper is showing the involvement of Orai1 in this process. They show initially that Orai1 is a primary molecular component of functional SOCE in podocytes. However, it is well known that Orai1 mediates Ca²⁺ entry in almost all cell types, so this finding alone is not of great significance. That Orai1 is a downstream effector of insulin signaling in podocytes is a novel finding and of some significance. However, the authors need to carefully answer the following questions on how this occurs.

Major Points:

(1) One big question is how the role of Orai1 differs from all the previously published findings on TRPC5 and TRPC6? It seems that the role of Orai1 just mimics many of the effects that were previously published on the TRPC channels. Does Orai1 overexpression or knockdown affect the role of TRPC channels? Does elimination of TRPC5 or TRPC6 alter the effects of insulin on Orai1? It seems curious and concerning that previously so much was published on the role of TRPCs in Ca²⁺ signaling in podocytes yet in the current paper there is nothing to address their role and any connection with the action of Orai1

(2) It is also curious why there is nothing in the current paper on the role of STIM proteins. Important to know is whether the activation of Orai1 channels in response to insulin is mediated by Ca²⁺ store-depletion and sensed by the STIM1 and/or STIM2 proteins. If the action of Orai1 is independent of STIM proteins this would definitely be notable. If the action of insulin is mediated by STIM proteins then it would be important to assess whether and how insulin receptors induce store-depletion and the subsequent activation of STIM proteins.

(3) It is unexpected that insulin would alter the turnover and surface expression of Orai1 channels. In almost all other studies on Orai1 channels, they are shown to be almost exclusively expressed in the PM. The authors show the role of the PI3-Akt pathway for altered turnover of Orai1 channels and this seems to mediate insulin-induced cell surface expression of Orai1. They show that without insulin, most of the Orai1 is expressed inside cells. But this seems to be at odds with many studies that show the almost exclusive PM expression of Orai1. What do the authors see with HEK cells? Is this expression pattern something specific for podocytes?

(4) There are clear genetic links between TRPC6 channels and podocyte function. Is there any genetic relationship between Orai1 and altered podocyte function? The authors mention a Taiwanese study, but did those findings have any correlation with the current paper?

Minor Point:

Although the paper is clearly written, there are numerous grammatical errors throughout the manuscript.

Reviewer #2 (Remarks to the Author):

The study by Kim et al. on insulin activation of Orai1 channels nicely demonstrates that Orai1 plays a major role in podocytes and that well-adjusted insulin levels are required to maintain glomerular filter integrity which clearly involves Orai1. This study definitely provides a good basis for novel therapeutic strategies.

I have few minor comments and questions especially regarding CRAC/Orai channels in podocytes:

- It is well known that CRAC channels are composed of STIM1 and Orai1. Those two components are sufficient to fully reconstitute CRAC channels, while additional regulators cannot be excluded. However, the authors mentioned nowhere in the results section how STIM1 is related to their Orai1 dependent effects in podocytes. Is STIM1 also expressed in podocytes? Is it possible that upon insulin application not only Orai1- but also STIM1-expression is enhanced? As STIM1 activates CRAC currents, it is likely that it also affects glomerular filter activity. Key experiments with a knock-down of STIM1 by siRNA would definitely strengthen the conclusions drawn.
- Is it possible to measure currents directly from podocytes utilizing the patch-clamp technique?
- It seems that Ca²⁺ entry in podocytes is about two-fold higher than in glomeruli together with a stronger inhibitory effect of 2-APB in the former cells. What's the reason for these differences?
- The authors used SKF and 2-APB to block SOCE in podocytes. GSK-7975 is a much more selective inhibitor of CRAC/Orai channels, which should be used in some key experiments to address CRAC/Orai channels in podocytes more specifically.
- Authors presented Orai1 activation via insulin in HEK cells as displayed in Fig. 2i, j. Was Orai1 expressed alone or together with STIM1 in HEK cells? I assume that STIM1 has been co-expressed which should be indicated in the legends or Results. Currents in the range of 240 pA/pF typically arise from absolute currents in the range of 5000 pA based on a single HEK cell capacitance of ~20pF. This seems to be quite high. With respect to the signaling pathways coupled to insulin, how do these pathways compare with each other in podocytes and HEK cells?
- Fig. 3d and Fig. 4e: Why are the relative Orai1 expression levels in response to insulin so much different?

Reviewer #3 (Remarks to the Author):

This manuscript by Cha et al. explores the mechanisms of store operated calcium channels in podocytes. Given the importance of calcium signaling in podocytes, the topic is of significant interest and therefore this study is timely.

However, the data appear in places inconsistent or inconclusive and at times they do not appear to support the main conclusions of the paper. There are significant scientific and technical inconsistencies. Overall these major concerns significantly dampened enthusiasm for this manuscript.

Specific comments and major concerns:

A general concern is that overall the authors pay little attention to a body of previous work by

many groups on TRPC channels in podocytes, for which there is also human genetic evidence (see TRPC6 mutations causing kidney disease, Winn et al, Science). They perform their experiments looking only at Orai without any studies to differentiate this from TRPC current. This is a particularly big concern given the fact that many of the tools used in this study (2APB for example) block both Orai and TRPC channels which makes it impossible to tell what is what. Experiments dedicated to resolving the contributions of Orai versus TRPC3,5,6,7 channels which are previously reported to be expressed and active in podocytes seem necessary for this study to be convincing.

Another general comment pertains to the over-reliance on calcium imaging and not enough patch clamp electrophysiology to assay channel function. And yet, no firm conclusions can be drawn about Orai channel activity without proper patch clamp studies in podocytes on mouse glomeruli, to show that the recorded calcium responses correspond to Orai channel activity (see Staruschenko et al, several papers). This is a very fundamental point and must be addressed throughout this manuscript. Patch clamp would also very much help resolve if these channels are indeed Orai1 versus TRPC channels, to the point above.

Another fundamental issue: As has been shown in the past, when calcium permeable channels are overexpressed in either cells or animals, they cause significant cellular toxicity (due to calcium overload) and therefore this is not a good system in which to study if a calcium permeable channel is implicated in disease. Basically, one causes the disease by simply perturbing the balance of calcium homeostasis, but this is not proof of the specific role of the channel in question in causing disease. Rather, these studies should be pursued with loss of function animal models (Orai knockout versus other Orai knockouts (Orai 2 and 3), versus TRPC6 knockouts, for example) to demonstrate that these animals are protected from disease.

A fundamental concept set forth in this manuscript is that insulin does not activate Orai1, but it only increases Orai1 channel translocation to the plasma membrane. If this is the case, then the authors' statements about "Orai1 activation by insulin" in Fig.2 and "insulin activates Orai1" in Fig.5 are indeed very misleading.

Insulin is known to increase TRPC channel expression in a similar mechanism (translocation to the plasma membrane) as shown here for Orai. Again, the authors must address whether insulin also affects TRPC channels or how it is possible that in their hands it only affects translocation of Orai. Experiments using either TRPC knock out animals or knock-down cells at the very least seems necessary to resolve these problems.

Fig3 a and b, there is almost no surface expression of Orai1 in mouse podocytes without insulin treatment. Yet they still see a large SOCE response, as shown in Fig1 a under these conditions. These two measurements appear to be contradicting each other. What is the explanation for this discrepancy?

Fig. 3: some of the compounds used (Brefeldin A, botulinum toxin, etc) have so many pleiotropic effects, it is hard to provide convincing evidence that the mechanisms described are in any way specific to Orai1. This is especially true when many of these interventions affect the actin cytoskeleton, which will definitely affect the localization of other channels including TRPC channels in podocytes.

Fig1. j, Orai1 does not appear to co-localize with synaptopodin, bringing into question whether Orai is expressed in podocytes, the cells of interest in this paper.

In contrast, in Fig6. b, overexpression of mcherry-Flag-Orai1 is perfectly co-localized with synaptopodin in rat glomeruli of podocytes. How is this reconciled with findings in Fig.1 and furthermore, how was expression limited to podocytes and not present in other glomerular cell types, how is this technically possible?

Moreover, this experiment itself causes concern due to artifacts that are often introduced by the overexpression of calcium-permeable channels in cells/animals. A more convincing experiment may be to knock-down or knock out Orai1 and study the insulin effect on glomerular podocytes in this setting.

Fig7. a, the authors should also show immunostaining results of Orai1 together with podocytes marker given concerns as above.

The Fig7 legend title is "Blockade of Orai1 ameliorates slit diaphragm dysfunction and proteinuria in hyperinsulinemic db/db mice". However, none of the data from Fig7 support this argument. Even if the overall expression of Orai1 is unregulated in db/db mouse glomeruli, there are no data offered to show whether Orai1 surface expression is increased (which we were told in Fig.2 is what is important for insulin signaling to Orai), moreover, there is no evidence as to whether the activity of Orai1 in db/db mouse glomeruli is increased. These experiments are absolutely essential to support the title of this Figure legend, and a more suitable legend in that case would be "Glomerular Orai1 activity is increased in db/db mice". Also the authors must really perform patch clamp experiments to obtain convincing Orai1 activity data.

The loss of function experiment would be required to gather convincing evidence that Orai1 mediated diabetic podocyte damage: either SOCE inhibitors or by CRISPR knockout, one would have to show that, in glomeruli, one can preserve podocyte numbers and reduce proteinuria using db/db mice. The authors would have to perform these experiments using db/db mice crossed to the Orai1 knockout mice and compare them to littermates in order to support their conclusions.

Fig. 8: the emergence of CsA as the drug that can reverse (ameliorate) the phenotype is puzzling. CsA has many effects, including most notably on NFAT, which has been shown to affect TRPC6 channel activity. Could this be an alternate explanation for these findings? Control experiments to address this possibility are missing. CsA also has effects on mitochondrial calcium, could this explain how mice are "rescued" Again controls are essential to determine this.

In conclusion, this study needs to be fundamentally rethought and redesigned, with improved controls and clearer consideration of alternate hypotheses to determine the role of Orai1 in podocyte pathobiology. The absence of human genetic evidence implicating Orai1 (in light of genetic evidence implicating other TRPC channels for instance), make the possibility of Orai1 contributing in a major way to disease less probable, and therefore even more efforts must be made to bolster its role in podocyte pathobiology.

We appreciate reviewers' helpful comments to improve the manuscript and have revised the manuscript fully accordingly. The revised text has changes from the original manuscript in red font to facilitate review.

Reviewer #1 (remarks to the Author):

Major Points:

1. One big question is how the role of Orai1 differs from all the previously published findings on TRPC5 and TRPC6? It seems that the role of Orai1 just mimics many of the effects that were previously published on the TRPC channels. Does Orai1 overexpression or knockdown affect the role of TRPC channels? Does elimination of TRPC5 or TRPC6 alter the effects of insulin on Orai1? It seems curious and concerning that previously so much was published on the role of TRPCs in Ca²⁺ signaling in podocytes yet in the current paper there is nothing to address their role and any connection with the action of Orai1

Our response:

Thank you for raising important issues. Orai1 is important in many cellular processes. TRPC and SOC channels are suggested as major Ca²⁺ influx mechanisms in epithelial cells and are downstream effectors of phospholipase C (PLC)-linked receptors. Our data demonstrate the distinct roles of Orai1 on actin dynamics and on RTK regulation. Hyperactivation of TRPC6 or TRPC5 in podocyte causes actin remodeling and proteinuria. A recent study demonstrated that TRPC5 and TRPC6 are antagonistic regulators of actin dynamics and cell motility in podocytes (Tian et al., Sci Signal, 3(145):ra77, 2010; reviewed by Greka and Mundel, Am J Soc Nephrol 22:1969-80, 2011). Under physiological condition, angiotensin II receptor stimulation activates both TRPC5 and TRPC6 with different phenotypes on actin dynamics and cellular behaviors. TRPC5-mediated Ca²⁺ influx decreases stress fiber assembly, thereby promoting cell migration, whereas TRPC6 activation promotes stress fiber formation and a contractile cell phenotype. In this study, Orai1-mediated Ca²⁺ influx shows decreased stress fiber formation and increased motility and contractility suggesting that the functional role of Orai1 on actin cytoskeleton may differ from all the previously published findings on TRPC5 and TRPC6 in podocyte. Ca²⁺ microdomains via TRPCs and Orai1 linking actin dynamics and cell migration remain to be clarified in future studies.

We have carried out additional studies whether Orai1 knockdown affects the role of TRPC5 and TRPC6 and elimination of TRPC5 or TRPC6 alters the insulin effect on Orai1. Knockdown Orai1 does not alter expression of TRPC5, TRPC6 and STIM1. Results are shown in new Supplemental Figure 1. On the other hand, we also found that knockdown of TRPC5 or TRPC6 affects neither Orai1/STIM1 expression nor Orai1-mediated SOCE. Moreover, the elimination of TRPC5 or TRPC6 did not alter Orai1 expression and insulin-

stimulated SOCE in cultured podocyte. Results are shown in revised Supplementary Figure 1 and 3. Insulin is known to stimulate cell surface expression of TRPC6 not TRPC5 in cultured podocytes. In this context, we further examined if elimination of TRPC6 alters the effect of insulin on Orai1-mediated SOCE using isolated glomeruli of TRPC6 knockout mice and found that insulin effect on SOCE is preserved in TRPC6 KO mice (see new Supplementary Figure3). Together, these results demonstrate that insulin may stimulate Orai1-mediated SOCE in podocytes independently of TRPC5 or TRPC6.

2. It is also curious why there is nothing in the current paper on the role of STIM proteins. Important to know is whether the activation of Orai1 channels in response to insulin is mediated by Ca²⁺ store-depletion and sensed by the STIM1 and/or STIM2 proteins. If the action of Orai1 is independent of STIM proteins this would definitely be notable. If the action of insulin is mediated by STIM proteins then it would be important to assess whether and how insulin receptors induce store-depletion and the subsequent activation of STIM proteins.

Our response:

We apologize that the previous manuscript did not examine the role of STIM1 protein. Yes, we have focused on Orai1 regulation in the former version of manuscript. STIM1 is critical for activation of CRAC channels and SOCE. We have carried out additional studies using siRNA STIM1 to regulate Orai1-mediated SOCE in podocyte. STIM1 is expressed in cultured podocyte and isolated glomeruli. Moreover, SOCE is markedly inhibited by STIM1 knockdown in mouse podocyte supporting that Orai1 activation is STIM1-dependent in podocyte (see new Supplemental Figure 1a-c). Of note, insulin treatment does not alter STIM1 expression (See new Figure 2g). Although insulin has no effect on STIM1 expression, however, it is also conceivable that insulin signaling can affect STIM1 activity via phosphorylation. Recent studies demonstrate that STIM1 phosphorylation by Erk1/2 upon IGF1 stimulation play a critical role for SOCE (Tomas-Martin et al., Cell Signal, 27:545-554, 2015; Pozo-Guisado et al., J Cell Sci, 126(Pt14):3170-80, 2013). Indeed, we also found that insulin activates Erk1/2 phosphorylation in cultured podocytes (data not shown in manuscript). In the present study, we have focused on insulin regulation of Orai1. We will examine whether STIM1 activity regulated by insulin and/or other RTKs is involved in podocyte Ca²⁺ signaling including Ora1s and TRPCs in the future studies.

3. It is unexpected that insulin would alter the turnover and surface expression of Orai1 channels. In almost all other studies on Orai1 channels, they are shown to be almost exclusively expressed in the PM. The authors show the role of the PI3-Akt pathway for altered turnover of Orai1 channels and this seems to mediate insulin-induced cell surface expression of Oria1. They show that without insulin, most of the Orai1 is expressed inside cells. But this seems to at odds with many studies that show the almost exclusive PM expression of Orai1. What do the authors see with HEK cells? Is this expression pattern

something specific for podocytes?

Our response:

Although significant knowledge has been accumulated regarding STIM1 subcellular distribution dynamics, little is known about the Orai1 trafficking. Orai1 is distributed diffusely in various cell types such as cystic fibrosis bronchial epithelial cells, oocyte and CHO cells (Balghi et al., FASEB J 25:4274-91, 2011; Yu et al., J Cell Biol 191:523-535, 2011). At rest, Orai1 actively recycles between an endosomal compartment and the cell membrane in oocytes and CHO cells. While a significant portion of total Orai1 is in cytosol at steady state, store-depletion shifts intracellular Orai1 to the plasma membrane (Yu et al., J Cell Biol 191:523-535, 2011). To examine the hypothesis that insulin receptor signaling affects Orai1 distribution and trafficking, podocytes were incubated in a serum-free medium before insulin treatment to exclude contamination of other growth factor effect (see Fig. 2a). We found that SOCE was reduced by serum deprivation suggesting that SOCE is regulated by serum growth factors at the steady state. Accordingly, while Orai1 localization is more diffusible in serum-reduced condition, cell surface localization of Orai1 is increased by insulin treatment supporting the steady-state recycling of Orai1 in mouse podocyte (see revised Fig. 3b). To clarify the location and redistribution of the Orai1 responded to insulin stimulation, HEK293 cells expressing Orai1 labeled with mCherry (mCherry-Orai1) are stimulated by insulin. Although mCherry-Orai1 is labeled at the plasma membrane in HEK293 cells, it is distributed diffusely. Insulin stimulation markedly shifts intracellular Orai1 to the plasma membrane as well as podocytes.

4. There are clear genetic links between TRPC6 channels and podocyte function. Is there any genetic relationship between Orai1 and altered podocyte function? The authors mention a Taiwanese study, but did those findings have any correlation with the current paper?

Our response:

Yes, it is well established that gain-of-function mutations of TRPC6 cause FSGS and proteinuric glomerular diseases. CRAC channelopathy is caused by loss- or gain-of-function mutations in *ORAI1* and *STIM1*. Gain-of-function mutations of in *ORAI1* and *STIM1* result in constitutive activation of CRAC channel or SOCE causing an overlapping spectrum of diseases including tubular aggregate myopathy (TAM), Stormorken syndrome and York platelet syndrome. However, it has not been reported yet whether disease phenotypes of CRAC channelopathy display kidney dysfunction such as proteinuria. While the *ORAI1* polymorphism rs12313273 is associated with hypercalcemia in Taiwanese CKD patients (Hwang et al., BioMed Res Int 2014:290863, 2014), nothing provides evidences that *ORAI1* mutation is related with kidney dysfunction including glomerular diseases in this paper. Therefore, to uncover the clinical features of genetic mutations of CRAC channels linked to kidney filter function awaits future investigation.

Minor Point:

Although the paper is clearly written, there are numerous grammatical errors throughout the manuscript.

Our response:

We corrected grammatical errors.

Reviewer #2 (remarks to the Author):

I have few minor comments and questions especially regarding CRAC/Orai channels in podocytes:

1. It is well known that CRAC channels are composed of STIM1 and Orai1. Those two components are sufficient to fully reconstitute CRAC channels, while additional regulators cannot be excluded. However, the authors mentioned nowhere in the results section how STIM1 is related to their Orai1 dependent effects in podocytes. Is STIM1 also expressed in podocytes? Is it possible that upon insulin application not only Orai1- but also STIM1-expression is enhanced? As STIM1 activates CRAC currents, it is likely that it also affects glomerular filter activity. Key experiments with a knock-down of STIM1 by siRNA would definitely strengthen the conclusions drawn.

Our response:

Thank you for raising this important issue. Yes, Orai1 and STIM1 are predominant molecular components of the CRAC channels and SOCE. Thus, we have carried out additional studies using siRNA STIM1 to regulate Orai1-mediated SOCE in podocyte. Please also see our response to comment #2 from reviewer1. We found that STIM1 is expressed in cultured podocyte and isolated glomeruli. SOCE is markedly blunted by STIM1 siRNA knockdown in mouse podocytes supporting the notion that Orai1 and STIM1 reconstitute CRAC channels in mouse podocytes (see new Supplemental Figure 1a-c). Additionally, STIM1 expression was not affected by insulin treatment (See new Figure 2g). While insulin has no effect on STIM1 expression, we don't completely exclude the possibility, if insulin signaling may affect STIM1 activity via phosphorylation. Indeed, recent studies demonstrated that STIM1 phosphorylation by Erk1/2 upon IGF1 stimulation plays a critical role for SOCE (Tomas-Martin et al., Cell Signal, 27:545-554, 2015; Pozo-Guisado et al., J Cell Sci, 126(Pt14):3170-80, 2013). We also found that insulin activates Erk1/2 phosphorylation in cultured podocytes (data not shown in MS). In the present study, we have focused on insulin regulation on Orai1.

We will examine whether STIM1 activity by insulin and/or other RTKs is implicated in podocyte Ca^{2+} signaling via regulating Orais and/or TRPCs in the next future studies.

2. Is it possible to measure currents directly from podocytes utilizing the patch-clamp technique?

Our response:

We fully agree that direct measurement of CRAC currents using patch clamp techniques may be helpful to figure out intrinsic channel properties of Orai1. There are two distinct approaches to measure CRAC current using patch-clamp techniques: single channel vs whole-cell dialyzed recordings. Firstly, CRAC channels have extremely small unitary conductance (10~25 fS estimated from noise analysis) compared to TRPCs (38~64 pS for TRPC5 vs 28~37 pS for TRPC6). We have tried to measure single channel current of Orai1; however, it has not been feasible due to very low conductance. Secondly, voltage-clamp technique for whole-cell recording is prone to errors including space-clamp errors which are due to electronic decay across distance with the space constant λ . Space-clamp errors can be reduced by recording from small cells with simple morphology, ideally spherical cells (Sonthemeier and Ransom, Patch Clamp Analysis Advanced Techniques, Human Press, 2002:35-67). Cultured podocytes are flattened shape cells with relatively high membrane capacitance indicative of a large cell size (range 90~200 pF for podocytes vs ~12 pF for HEK293 cells). This feature creates space-clamp errors and overcompensation problem. In the present study, we have been eager to measure whole-cell CRAC and TRPC6 current in cultured podocytes; however, it has not been feasible due to space-clamp error and overcompensation problem. Thus, we examined the intrinsic channel properties of CRAC currents regulated by insulin using heterologous expression in HEK293 cells that are much smaller in size (mean membrane capacitance is ~12 pF).

3. It seems that Ca^{2+} entry in podocytes is about two-fold higher than in glomeruli together with a stronger inhibitory effect of 2-APB in the former cells. What's the reason for these differences?

Our response:

Yes, in previous version of manuscript, we showed that Ca^{2+} entry and 2-APB effects are much higher in mouse podocytes compared to that of isolated rat glomeruli. We suggest several possibilities for different SOCE and inhibitor action; for example, single podocyte vs isolated glomeruli or mouse vs rat. Isolated glomeruli contains a lots of podocytes which are connected each other in the glomeruli. It is conceivable that Fura-2 and inhibitors may be more diffusible to the neighboring podocytes suggesting that Ca^{2+} influx and pharmacological inhibition are more efficient in podocytes than in glomeruli. We also found

that SOCE in mouse isolated glomeruli is much higher than in rat glomeruli. We carried out additional studies using isolated mouse glomeruli and GSK-7975A, a selective inhibitor of Orai channels (see revised Fig. 1a-e and Fig. 2b,c). These additional studies suggest that Ca^{2+} entry and inhibitor effects may vary depending on tissue or cell types and/or species.

4. *The authors used SKF and 2-APB to block SOCE in podocytes. GSK-7975 is a much more selective inhibitor of CRAC/Orai channels, which should be used in some key experiments to address CRAC/Orai channels in podocytes more specifically.*

Our response:

This is actually a very valuable suggestion. Yes, SKF96365 and 2-APB affect multiple channels including TRPCs and OraIs. GSK-7975A, a pyrazole derivative, is a much selective inhibitor of Orai1 and Orai3 current. Following the reviewer's suggestion, we have redesigned pharmacological experiments and have carried out additional studies using GSK-7975A in *in vitro*, isolated glomeruli and *in vivo* experiments throughout the manuscript, accordingly. Effects of GSK-7975A are similar to that of 2-APB. Notably, SKF96365 not only functions as channel blockers but also inhibits Akt signaling in A7r5 vascular smooth cells (Park et al., *Biochim Biophys Acta.* 1813(12)2157-64, 2011). To be cautious, we exclude the data about SKF96365 effect on SOCE in podocytes in previous manuscript. Revised results are shown in Figure 1, 2,7, and 8 and Supplemental Figure 2.

5. *Authors presented Orai1 activation via insulin in HEK cells as displayed in Fig. 2i, j. Was Orai1 expressed alone or together with STIM1 in HEK cells? I assume that STIM1 has been co-expressed which should be indicated in the legends or Results. Currents in the range of 240 pA/pF typically arise from absolute currents in the rage of 5000 pA based on a single HEK cell capacitance of ~20pF. This seems to be quite high. With respect to the signaling pathways coupled to insulin, how do these pathways compare with each other in podocytes and HEK cells?*

Our response:

We apologize that previous manuscript did not accurately provide transfection information and mislabeling. mCherry-tagged Orai1 was cotransfected with YFP-tagged STIM1 in HEK293 cells. We indicated in the revised figure legends and methods. Yes, as noted by the reviewer, absolute current values are too high. There was mislabeling in Fig. 2i, j in previous manuscript (pA → pA/pF). We have carefully checked and re-analyzed the raw data used in previous manuscript and have repeated patch-clamp experiments using HEK293 cells expressing Orai1 and STIM1 to be sure. The mean capacitance of HEK293 cells is ~12 pF. Representative I-V relationship curves in the presence or absence of insulin treatment are shown in revised Fig. 2h. Summary of the results of normalized CRAC current density

(pA/pF) at -100 mV is shown in revised Fig. 2i. To be cautious, we corrected labeling in revised Fig. 2h,i.

5. Fig. 3d and Fig. 4e: Why are the relative Orai1 expression levels in response to insulin so much different?

Our response:

We thank the reviewer for noticing this discrepancy and apologize for being unclear. In Fig. 4e in former manuscript, the western blot images showing quite low surface Orai1 expression was due to less exposure time. Hence, relative insulin-mediated surface expression of Orai1 was quite high in Fig. 4e of former manuscript. We have carefully checked figures with different expose time. To be sure, we have repeated experiments, and have replaced the pictures with longer exposure time showing functional surface abundance of Orai1 (see revised Fig. 3 and 4). To be able to compare with certainty, we repeated experiments for biotinylation assay and Ca²⁺ imaging with same condition side by side at the same day.

Reviewer #3 (remarks to the Author):

Specific comments and major concerns:

1. A general concern is that overall the authors pay little attention to a body of previous work by many groups on TRPC channels in podocytes, for which there is also human genetic evidence (see TRPC6 mutations causing kidney disease, Winn et al, Science). They perform their experiments looking only at Orai without any studies to differentiate this from TRPC current. This is a particularly big concern given the fact that many of the tools used in this study (2APB for example) block both Orai and TRPC channels which makes it impossible to tell what is what. Experiments dedicated to resolving the contributions of Orai versus TRPC3,5,6,7 channels which are previously reported to be expressed and active in podocytes seem necessary for this study to be convincing.

Our response:

We thank the reviewer for raising important issues. Yes, it is well-established that TRPC5 and TRPC6 are expressed in podocytes and play a critical role in intracellular Ca²⁺ homeostasis and actin dynamics. In the previous version of manuscript, we have focused on Orai1 regulation. Following the reviewer suggestions, as in the response to comment1 for reviewer#1, we have carried out additional studies how and if the role of Orai1 differs from

that of TRPC5 and TRPC6. Hence, we examined whether Orai1 knockdown affects TRPC5 and TRPC6. Knockdown Orai1 did not alter the expression of TRPC5, TRPC6 and STIM1 (Result is shown in new Supplemental Figure 1). Next, we also examined whether elimination of TRPC5 or TRPC6 alters the insulin effect on Orai1. We found that knockdown of TRPC5 or TRPC6 affects neither Orai1/STIM expression nor Orai1-mediated SOCE and insulin effect on Orai1 in cultured podocytes. Results are shown in revised Supplementary Figure 1 and 3. Additionally, insulin is known to stimulate cell surface expression of TRPC6 not TRPC5 in cultured podocytes indicating that both channels are differently regulated by RTKs (Kim et al., Am J Physiol Renal Physiol. 302:F298-307, 2012). In this context, we further examined whether elimination of TRPC6 alters the insulin effects on Orai1-mediated SOCE using isolated glomeruli of TRPC6 knockout mice and found that insulin effect on SOCE is preserved in TRPC6 KO mice (see Supplementary Figure3). Together, these results demonstrate that insulin stimulates Orai1-mediated SOCE in podocyte independently of TRPC5 or TRPC6.

We agree and thank for constructive criticism. SKF96365 and 2-APB affect multiple channels including TRPCs and Orais. While 2-APB inhibits Orai1 in a STIM1-dependent mechanism at high concentration, 2-APB directly activates Orai3 independently of ER Ca²⁺ depletion and STIM1 (Prakriya and Lewis, Physiol Rev. 95:1383-436, 2015). A pyrazole derivative, GSK-7975A is a much selective inhibitor of Orai1 and Orai3 currents (Derler et al., Cell Calcium 53:139-51, 2013). Per suggestion, we have redesigned pharmacological experiments and have carried out additional studies using GSK-7975A in *in vitro*, isolated glomeruli and *in vivo* experiments (see new Figures; Fig. 1, 2, 7, and 8 and Supplemental Figure2). SOCE was blocked by inhibitors, 2-APB and GSK-7975A demonstrating that Orai1 is a major component of SOCE in podocytes. Notably, SKF96365 not only functions as channels blockers but also inhibits Akt signaling in A7r5 vascular smooth cells (Park et al., Biochim Biophys Acta. 1813(12)2157-64, 2011). To be cautious, we exclude the data from former manuscript about SKF96365 effect on SOCE in podocytes.

2. Another general comment pertains to the over-reliance on calcium imaging and not enough patch clamp electrophysiology to assay channel function. And yet, no firm conclusions can be drawn about Orai channel activity without proper patch clamp studies in podocytes on mouse glomeruli, to show that the recorded calcium responses correspond to Orai channel activity (see Staruschenko et al, several papers). This is a very fundamental point and must be addressed throughout this manuscript. Patch clamp would also very much help resolve if these channels are indeed Orai1 versus TRPC channels, to the point above.

Our response:

We agree with the point raised by the reviewer. Yes, direct measurement of CRAC currents using patch clamp techniques may be helpful to figure out intrinsic channel properties of Orai1. As in the response to comment2 for reviewer#2, there are two distinct approaches to measure CRAC current using patch-clamp techniques: single-channel vs whole-cell dialyzed

recordings. Staruschenko group recorded single channel TRPC6 currents but not whole-cell current in isolated glomeruli in their several papers. CRAC channels have extremely small unitary conductance (10~25 fS estimated from noise analysis) compared to TRPCs (38~64 pS for TRPC5 vs 28~37 pS for TRPC6). We have tried to measure single channel current of CRAC channel; however, single-channel recording was not successful due to small unitary conductance. As in the response to reviewer#2, voltage-clamp technique is prone to errors including space-clamp errors which are due to electronic decay across distance with the space constant λ . Space-clamp errors can be reduced by recording from small cells with simple morphology, ideally spherical cells (Sonthemeier and Ransom, Patch Clamp Analysis Advanced Techniques, Human Press, 2002:35-67). Cultured podocytes are flattened shape cells with relatively high membrane capacitance indicative of a large cell size (90~200 pF for podocytes vs ~12 pF for HEK293 cells). This feature has created space-clamp errors and overcompensation phenomenon. In the present and previous our study (Kim et al., Am J Soc Nephrol, 28(1):140-51, 2017), we have been eager to measure whole-cell current for CRAC and TRPC6 in cultured podocytes, however, it has not been feasible due to space-clamp errors. Thus, we examined the intrinsic channel properties of CRAC currents regulated by insulin using heterologous expression in HEK293 cells that are much smaller in size (mean membrane capacitance is ~12 pF).

3. Another fundamental issue: As has been shown in the past, when calcium permeable channels are overexpressed in either cells or animals, they cause significant cellular toxicity (due to calcium overload) and therefore this is not a good system in which to study if a calcium permeable channel is implicated in disease. Basically, one causes the disease by simply perturbing the balance of calcium homeostasis, but this is not proof of the specific role of the channel in question in causing disease. Rather, these studies should be pursued with loss of function animal models (Orai knockout versus other Orai knockouts (Orai 2 and 3), versus TRPC6 knockouts, for example) to demonstrate that these animals are protected from disease.

Our response:

Yes, Ca^{2+} overload can cause significant cellular toxicity. The goal of our study is to investigate whether overexpression of Orai1 is a culprit of albuminuria and insulin signaling targeting Orai1 contributes to aggravation of albuminuria. Compared to the control mice, the mice overexpressing *Orai1* transgene developed significant albuminuria (Fig. 6c) and foot process effacement (Fig. 6d,e), indicating that *Orai1* overexpression is sufficient to induce slit diaphragm dysfunction and albuminuria. Moreover, insulin aggravates proteinuria in Orai1-overexpressing mice. To clarify if SOCE is implicated in albuminuria, we have carried out additional studies using 2-APB administration to inhibit Orai1-mediated Ca^{2+} influx in mice overexpressing *Orai1* transgene. Basal and insulin-stimulated albuminuria was blunted by administration of 2-APB (see revised Fig. 6f-h). In addition, as shown in revised Fig. 7 and 8, Orai1 is overexpressed in cultured podocyte and in the hyperinsulinemic *db/db* mice supporting Orai1 overexpression causing filter dysfunction *in vivo*. We have carried out

additional studies demonstrating that albuminuria and foot process effacement were rescued by GSK7975A or 2-APB administration in *db/db* mice (Revised Fig. 8 and Supplemental Fig. 6). As in the response to comment1 for reviewer#1 and above, it is known that insulin activates TRPC6 not TRPC5 via promoting cell surface abundance in cultured podocytes indicating that both channels are differently regulated by RTKs (Kim et al., Am J Physiol Renal Physiol, 302:F298-307, 2012). In this study, we show that insulin upregulates Orai1-mediated Ca^{2+} influx in TRPC6 knockout mice supporting the notion that insulin signaling mediates Ca^{2+} influx via Orai1 independently of TRPC6 in podocytes (see new Supplementary Figure3). Our study expands the list of Ca^{2+} -permeable channels potentially regulating actin dynamics and filter integrity in podocytes. Dissecting microdomain of Ca^{2+} signaling via TRPCs and Orai1 linking actin dynamics remain to be clarified in future studies.

4. A fundamental concept set forth in this manuscript is that insulin does not activate Orai1, but it only increases Orai1 channel translocation to the plasma membrane. If this is the case, then the authors' statements about "Orai1 activation by insulin" in Fig.2 and "insulin activates Orai1" in Fig.5 are indeed very misleading. Insulin is known to increase TRPC channel expression in a similar mechanism (translocation to the plasma membrane) as shown here for Orai. Again, the authors must address whether insulin also affects TRPC channels or how it is possible that in their hands it only affects translocation of Orai. Experiments using either TRPC knock out animals or knock-down cells at the very least seems necessary to resolve these problems.

Our response:

We agree and have changed the statements to “Insulin upregulates Orai1 ...” in Fig. 2 and 4 and “Insulin stimulation of Orai1...” in Fig. 5 in the revised manuscript. Yes, insulin can also affect TRPC channels. As in the response to comment1 for reviewer#1 and to point 1 above, it is reported that insulin stimulates TRPC6 not TRPC5 via promoting cell surface abundance in a NADPH oxidase-dependent mechanism in cultured podocytes suggesting that both channels are differently regulated by RTKs (Kim et al., Am J Physiol Renal Physiol, 302:F298-307, 2012). We have reported previously that PI3K-Akt pathway stimulates cell surface abundance of TRPC6 in podocytes and cardiac myocytes (Kim et al., Am J Soc Nephrol 28(1):140-51, 2017; Xie et al., Nat Commun, 3:1238, 2012). We have carried out additional studies whether Orai1 knockdown affects the role of TRPC5 and TRPC6 and elimination of TRPC5 or TRPC6 alters the insulin effect on Orai1. Knockdown Orai1 do not alter expression of TRPC5, TRPC6 and STIM1 (Result is shown in revised Supplemental Figure 1). Additionally, knockdown of TRPC5 or TRPC6 affects neither Orai1/STIM expression nor Orai1-mediated SOCE and insulin effect on Orai1 in cultured podocyte. Results are shown in Supplementary Figure 1 and 3. As mentioned above, insulin is known to stimulate cell surface expression of TRPC6 not TRPC5 in cultured podocytes (Kim et al., Am J Physiol Renal Physiol. 302:F298-307, 2012). In this context, we further examined if elimination of TRPC6 alters the insulin effects on Orai1-mediated SOCE using isolated glomeruli of TRPC6 knockout mice and found that insulin effect on SOCE is not altered in

TRPC6 KO mice (see Supplementary Figure3). Together, these results demonstrate that insulin may stimulate Orai1-mediated SOCE in podocytes independently of TRPC5 or TRPC6.

5. Fig3 a and b, there is almost no surface expression of Orai1 in mouse podocytes without insulin treatment. Yet they still see a large SOCE response, as shown in Fig1 a under these conditions. These two measurements appear to be contradicting each other. What is the explanation for this discrepancy?

Our response:

In Fig. 3a and b of the previous version of manuscript, the images showing quite low surface expression Orai1 was due to less exposure time. To be sure, we have repeated experiments, and have replaced the pictures with longer exposure time showing functional surface abundance of Orai1 (see revised Fig. 3a and b). To be able to compare with certainty, we repeated experiments for biotinylation assay and Ca^{2+} imaging with the same condition side by side at the same day.

To examine the hypothesis that serum growth factor signaling by targeting Orai1 affects podocyte behavior, podocytes were incubated in a serum-free medium before treatment of indicated serum or growth factors to exclude contamination of other growth factor effect (see Fig. 2a). We found that SOCE was reduced by serum deprivation suggesting that SOCE is regulated by serum growth factors at the steady state. We measured SOCE from the cells incubated in a serum-containing regular culture medium in Fig. 1. In Fig. 2a and following experiments to examine insulin effects throughout the paper, cells or isolated glomeruli were incubated in a serum-free medium for ~16 hr before insulin treatment for indicated duration. We indicated experiment condition in the figure legend to be cautious.

6. Fig. 3: some of the compounds used (Brefeldin A, botulinum toxin, etc) have so many pleiotropic effects, it is hard to provide convincing evidence that the mechanisms described are in any way specific to Orai1. This is especially true when many of these interventions affect the actin cytoskeleton, which will definitely affect the localization of other channels including TRPC channels in podocytes.

Our response:

Yes, we agree with the point raised by reviewer that Brefeldin A (BFA) and BoNT have pleiotropic effects. BFA is known to inhibit protein transport from the ER to the Golgi. BFA inhibits not only insulin-stimulated SOCE (see Fig. 3d) but also cell surface abundance of Orai1 (data not shown) demonstrating that BFA prevents Orai1 delivery to plasma membrane. On the other hand, we previously reported that BFA treatment blocks insulin-stimulated exocytosis of K_{ATP} channels and Akt phosphorylation in INS-1E cells (Xu et al., Biochem Biophys Res Commun. 468:752-757). For this reason, we examined whether BFA

blunts insulin-mediated Akt phosphorylation in podocytes. Similar to our previous finding in INS-1E cells, insulin-stimulated Akt phosphorylation was inhibited by BFA treatment. This result raises at least two possibilities for BFA action on insulin regulation of Orai1; BFA inhibits (1) Orai1 protein delivery to plasma membrane, or (2) Akt phosphorylation. To be cautious, we have mainly focused on VAMP2-associated exocytosis pathways of Orai1 in revised manuscript. We carried out additional experiments that insulin-induced surface expression was blunted by TeNT treatment. This data supports that insulin increase VAMP-2 associated exocytosis of Orai1 (see revised Fig. 3e,f). Of note, we have carried out additional studies whether the mechanism of insulin promoting Orai1 delivery to plasma membrane is similar to that described in previous reports on insulin-stimulated GLUT4 exocytosis. We found that Akt substrate AS160 (also known as TBC1D4) is involved in Orai1 exocytosis at basal steady state and under insulin stimulation. Results are shown in revised Figure 4f-h and discussed in details in the "Discussion". We have previously reported that serum growth factors stimulate VAMP2-associated TRPC6 exocytosis via PI3K-Akt pathway in podocytes (Kim et al., *Am J Soc Nephrol* 28(1):140-51). Previous findings with this study demonstrate that RTKs-PI3K-Akt signaling cascade may be a common regulator of exocytic pathway for Orai1 and TRPC6, at least in part, in podocytes.

7. Fig1. j, Orai1 does not appear to co-localize with synaptopodin, bringing into question whether Orai is expressed in podocytes, the cells of interest in this paper. In contrast, in Fig6. b, overexpression of mcherry-Flag-Orai1 is perfectly co-localized with synaptopodin in rat glomeruli of podocytes. How is this reconciled with findings in Fig.1 and furthermore, how was expression limited to podocytes and not present in other glomerular cell types, how is this technically possible?

Moreover, this experiment itself causes concern due to artifacts that are often introduced by the overexpression of calcium-permeable channels in cells/animals. A more convincing experiment may be to knock-down or knock out Orai1 and study the insulin effect on glomerular podocytes in this setting.

Our response:

In response to this valid point, we have repeated experiments of the co-localization of synaptopodin and Orai1 in the kidney sections and carried out additional experiments using isolated mouse glomeruli. New results are shown in Fig. 1j and k. Our results of Orai1 expression in podocytes are overwhelming, including 1) immunofluorescence imaging in kidney tissues, isolated glomeruli, and cultured mouse podocytes, 2) immunohistochemistry in mouse kidney tissues, and 3) western blot analysis and gene expression (real-time PCR) in cultured mouse podocyte cell line.

In Fig. 6, we employed a mouse model by Orai1 transgene delivery and the recombinant mCherry-Flag-tagged Orai1 protein was detected by mCherry fluorescence after gene delivery. In our previous study for TRPC6 gene delivery in mice, we showed that TRPC6

transgene was mostly co-localized with synaptopodin (Kim et al., Am J Soc Nephrol 28(1):140-51, 2017). While endogenous Orai1 is detected in podocytes as well as in mesangial cells and tubular cells (Fig. 1k), the exogenous Orai1 by gene delivery is mostly co-localized with synaptopodin (Fig. 6). The effect conferred by Orai1 in podocyte is very clear because Orai1 transgene-delivered mice have >2~3-fold increase in albuminuria compared to mCherry-vector control group. To be sure, we have carried out additional experiments that insulin-stimulated SOCE and synaptopodin dissolution were rescued by 2-APB administration in *Orai1*-overexpressing mice (revised Fig. 6f-h).

8. Fig7. a, the authors should also show immunostaining results of Orai1 together with podocytes marker given concerns as above.

Our response:

As suggested by the reviewer, we have done additional experiment showing colocalization of Orai1 and synaptopodin in the kidney sections from *db/db* and *db/m* mice. New results are shown in Fig. 7d.

9. The Fig7 legend title is "Blockade of Orai1 ameliorates slit diaphragm dysfunction and proteinuria in hyperinsulinemic *db/db* mice". However, none of the data from Fig7 support this argument. Even if the overall expression of Orai1 is unregulated in *db/db* mouse glomeruli, there are no data offered to show whether Orai1 surface expression is increased (which we were told in Fig.2 is what is important for insulin signaling to Orai), moreover, there is no evidence as to whether the activity of Orai1 in *db/db* mouse glomeruli is increased. These experiments are absolutely essential to support the title of this Figure legend, and a more suitable legend in that case would be "Glomerular Orai1 activity is increased in *db/db* mice". Also the authors must really perform patch clamp experiments to obtain convincing Orai1 activity data.

Our response:

Yes, we agree and apologize for being unclear. To be cautious, we have changed the Fig. 7 legend title to "Orai1 is upregulated via SGK1-dependent mechanism in hyperinsulinemic *db/db* mice". To examine Orai1 activity, we have performed additional Ca^{2+} imaging experiments using isolated glomeruli from *db/db* and *db/m* mice. SOCE was increased in *db/db* mice compared to the control mice supporting that functional SOCE is upregulated in podocytes of the *db/db* mice (see new Fig. 7a and b).

10. The loss of function experiment would be required to gather convincing evidence that Orai1 mediated diabetic podocyte damage: either SOCE inhibitors or by CRISPR knockout, one would have to show that, in glomeruli, one can preserve podocyte numbers and reduce

proteinuria using *db/db* mice. The authors would have to perform these experiments using *db/db* mice crossed to the *Orai1* knockout mice and compare them to littermates in order to support their conclusions.

Our response:

We thank the reviewer for this constructive suggestion. We have carried out additional rescue experiments using SOCE inhibitors, GSK-7975A and 2-APB in *db/db* mice. Increased albuminuria, synaptopodin dissolution, and foot process effacement and altered GBM thickness in *db/db* mice were rescued by administration of GSK-7975A or 2-APB supporting the notion that inhibition of *Orai1* hyperactivation ameliorates kidney filter dysfunction causing albuminuria in *db/db* mice. New results are shown in Fig. 8 and Supplemental Fig. 6.

11. Fig. 8: the emergence of CsA as the drug that can reverse (ameliorate) the phenotype is puzzling. CsA has many effects, including most notably on NFAT, which has been shown to affect TRPC6 channel activity. Could this be an alternate explanation for these findings? Control experiments to address this possibility are missing. CsA also has effects on mitochondrial calcium, could this explain how mice are “rescued” Again controls are essential to determine this.

Our response:

The reviewer raises a very important issue. Accumulating evidences suggest that podocyte is a direct target for antiproteinuric agent, CsA. Yes, CsA blunts the Ca^{2+} -calcineurin-NFAT signaling pathway. NFAT promotes *Orai1* transcription as well as TRPC6. CsA has been shown to ameliorate proteinuric glomerular diseases, both via NFAT-mediated transcriptional regulation and by protecting synaptopodin degradation. We show that insulin transiently increases *Orai1*-mediated Ca^{2+} influx followed by increased podocyte actin dynamics and synaptopodin dissolution. The SOCE stimulation by insulin was detected after 10 min incubation and reached maximal stimulation at 1 hr (Supplemental Fig. 2e,f). Moreover, mRNA expression of *Orai1* is not altered in hyperinsulinemic *db/db* mice and by long-term treatment of insulin *in vitro* suggesting that the stability of *Orai1* protein may be increased by insulin treatment. Of note, because of the short timescale and mRNA expression, our study supports the notion that CsA ameliorates albuminuria via mainly preventing synaptopodin degradation and actin remodeling by *Orai1*-mediated Ca^{2+} influx rather than NFAT-mediated transcriptional regulation of *Orai1*.

Multiple studies have demonstrated that mitochondria are key regulator of *Orai1* activity. Yes, CsA also blocks mitochondrial permeability transition pore (PTP) opening induced by pathological mitochondrial Ca^{2+} overload. Prevention of PT-mediated apoptosis could be another beneficial action of CsA against glomerular diseases.

12. In conclusion, this study needs to be fundamentally rethought and redesigned, with

improved controls and clearer consideration of alternate hypotheses to determine the role of Orai1 in podocyte pathobiology. The absence of human genetic evidence implicating Orai1 (in light of genetic evidence implicating other TRPC channels for instance), make the possibility of Orai1 contributing in a major way to disease less probable, and therefore even more efforts must be made to bolster its role in podocyte pathobiology.

Our response:

We thank the reviewer for the constructive suggestion. We agree that the previous version of manuscript had weak points such as lacking data on the role of important regulators for podocyte Ca²⁺ signaling including TRPC5, TRPC6 and STIM1. These comments by reviewer allow us to clarify and enforced the impact of this study. As suggested by the reviewer, we have now redesigned and carried out additional experiments and revised manuscript to improve the overall clarification.

Reviewers' comments:

Reviewer #1 has commented on your revised manuscript only confidentially and this is the reason you will not see his comments. He thinks the manuscript has been partially improved but he is not convince the work would have the level of advance we seek in a Nature Communications publications. Importantly, he also noticed that several of the reviewer #3 concerns were not addresses.

Reviewer #2 (Remarks to the Author):

no further comments

Reviewer #3 (Remarks to the Author):

In this revised manuscript, the authors have added some additional data to support their conclusions. However, I am afraid that none of the major concerns of this reviewer were adequately addressed, and in fact none of the key suggested experiments were performed.

Major concerns not addressed:

1. There are no patch clamp experiments in podocytes or intact glomeruli, which is an absolute requirement for publication of this manuscript as this is the standard in the field. Technical problems are stated as the reason for not doing these experiments, but the space clamping problems and other technical difficulties as raised by the authors here have been overcome by other groups and it is not a reason not to perform these studies. HEK studies are not a suitable alternative to podocyte or glomeruli recordings since the signaling pathways invoked here are podocyte specific. Additionally, single channel recordings in podocytes or intact glomeruli are not hampered by space clamping problems because a patch of membrane can be excised for single channel recordings in the inside-out or outside-out configuration, so there is really no technical reason not to do this work. Patch clamp electrophysiology in podocytes and intact glomeruli is possible and should be performed for this manuscript to be credible.
2. The requirement to show that loss of function of Orai is protective just as gain of function (transgenics/overexpression) causes disease is also a requirement for publication of this manuscript. The reason is again critically important and simple: the transgenic overexpression of a calcium permeable channel can cause cell injury/death because of calcium overload independent of any signaling events or specific events related to the biology of this cell, meaning the overexpression experiments on which this manuscript relies are greatly prone to artifacts and can mislead the investigators to erroneous conclusions. For example, if one were to overexpress a voltage gated calcium channel in a podocyte, that may also lead to cell injury and proteinuria due to calcium overload/podocyte death, but there is no evidence that voltage gated calcium channels are meaningfully involved in the biology of this cell type. Therefore, the only way to control and secure against this possible artifact is to perform loss of function studies to show that podocyte specific Orai knockout mice are protected from podocyte injury/proteinuria (as caused by a number of agents such as protamine sulfate, puromycin aminonucleoside, Adriamycin, high glucose diet etc). This must be done for the manuscript to be credible.
3. Consideration of the effects of other TRPC channels was also raised as a major concern by this reviewer. The authors have done studies to show that Orai knockdown does not alter the expression of TRPC5 and TRPC6 channels, but since ion channel activity rather than expression is what is critically important here, we return to point 1, which is that electrophysiology studies in podocytes looking at whether TRPC5, TRPC6 or other TRPC or Orai currents are affected by detailed pharmacology and genetic studies (using all available drugs and using knockdowns for each of those channels) is the only way to separate between these different channels, and support the novelty of the findings in this manuscript. The fact that insulin mediates calcium influx in TRPC6 knockout mice, which is offered as a new study in support of the conclusion that this is due

to Orai channel activity is not conclusive because it does not address any possible contributions from TRPC5 or other calcium permeable channels and it relies on drugs that are not specific for Orai.

Additional major ongoing concerns:

4. The pharmacologic studies (as in my previous point 6) remain problematic as the drugs used are non-specific and can have vast effects on the actin cytoskeleton. It is impossible to separate cause and effect here, despite the new studies (with a focus on VAMP2), which however have the same limitations because they rely on the same non-specific drugs. Genetic studies and electrophysiology to bolster these conclusions would be required for this part of the manuscript to be convincing. For example, since there are major concerns about the surface trafficking studies, it would be convincing to record Orai channel activity in podocytes under these conditions to conclusively demonstrate that Orai channels are absent or present in the plasma membrane. In the absence of this, the focus on exocytosis/trafficking as a mechanism for the regulation of Orai is not conclusive and contradicts previously reported work, as also noted by another reviewer.

5. The authors point out correctly that endogenous Orai is not only detected in podocytes but also in mesangial cells and tubular cells (Fig. 1 and new Fig 7d as well) (i.e. it is present throughout the entire kidney). Since insulin can also act on all these different cell types in the kidney, how can we be sure of the specificity of the insulin-Orai pathway in podocytes as claimed in this study? Again, the loss of function experiment is critical here: the podocyte specific deletion of Orai would be the only way to be convincing, if one can show that the deletion of Orai in podocytes is protective of podocyte injury/proteinuria.

6. The new data in Fig. 7 (to address point 9 from previous review) are not sufficient to support the conclusion of this study in db/db mice. Again, only calcium imaging is employed to address what the authors call SOCE (store operate calcium entry) but these studies can not demonstrate that this is specifically related to Orai channel activity. Again, patch clamp electrophysiology in intact db/db mouse glomeruli and in podocyte-specific Orai knockout crossed to db/db mice is critically important to demonstrate this point here.

7. In response to point 10 from previous review, the authors have not performed any genetic studies as suggested, and instead have relied again on drugs that they have already acknowledged are non-specific for Orai or even for SOCE. The only convincing way to address the issue of lack of drug specificity is to use Orai knockout mice.

8. In response to point 11 from previous review, the authors suggest that the short time scale of their observations is sufficient to support the conclusion that that NFAT-mediated transcriptional regulation of Orai (or other channels as previously published for TRPC6) is not relevant in their studies. This is an erroneous conclusion, as transcriptional events are well known to occur on short time scales. Further, rescue of synaptopodin degradation by calcineurin has been previously shown to be the result of inhibiting the effect of Cathepsin L in podocytes (Faul et al, Nat Med, 2008) and more recently as the result of blocking Vav2 activity in podocytes (Buvall et al, JASN, 2017). None of these pathways (as well as the mitochondrial pathway) require or implicate Orai activity and therefore the response offered to this point does not actually address the point.

Unfortunately, overall the revised manuscript does not address the major concerns raised and therefore it is not convincing as to the role of Orai in podocyte pathobiology.

Point-by-point response for NCOMMS-16-25536A

Response to Reviewer's Comments

Thank you for your consideration of our manuscript and helpful guidance on improving its presentation. We have tried to revise the manuscript in accordance with your guidance and the reviewers' comments. Details of our revision are described in our responses to reviewers' comments.

The current manuscript contains additional results from experiments suggested by the reviewers, including experimental data from tissue-specific knockout (podocyte-specific deletion of *Orai1*) mice, comparative data analysis from control (*db/m*) and *db/db* mice, and CRAC current recording from podocytes. We believe that these addition results strengthen the reliability of our hypothesis and extend the applicability of our take-home messages.

Reviewers' comments:

Reviewer #1 has commented on your revised manuscript only confidentially and this is the reason you will not see his comments. He thinks the manuscript has been partially improved but he is not convince the work would have the level of advance we seek in a Nature Communications publications. Importantly, he aslo noticed that several of the reviewer #3 concerns were not addresses.

Reviewer #2 (Remarks to the Author):

no further comments

Reviewer #3 (Remarks to the Author):

In this revised manuscript, the authors have added some additional data to support their conclusions. However, I am afraid that none of the major concerns of this reviewer were adequately addressed, and in fact none of the key suggested experiments were performed.

Major concerns not addressed:

1. *There are no patch clamp experiments in podocytes or intact glomeruli, which is an absolute requirement for publication of this manuscript as this is the standard in the field. Technical problems are stated as the reason for not doing these experiments, but the space clamping problems and other technical difficulties as raised by the authors here have been overcome*

by other groups and it is not a reason not to perform these studies. HEK studies are not a suitable alternative to podocyte or glomeruli recordings since the signaling pathways invoked here are podocyte specific. Additionally, single channel recordings in podocytes or intact glomeruli are not hampered by space clamping problems because a patch of membrane can be excised for single channel recordings in the inside-out or outside-out configuration, so there is really no technical reason not to do this work. Patch clamp electrophysiology in podocytes and intact glomeruli is possible and should be performed for this manuscript to be credible.

Our response:

We agree with the point raised by the reviewer. Direct measurement of Ca^{2+} release-activated Ca^{2+} (CRAC) currents using patch clamp techniques is required for identifying the intrinsic channel properties of Orai1 in podocytes. As mentioned previously, in flat and large cells, the voltage-clamp technique is prone to space-clamp errors, which are due to electronic decay across distance with the space constant λ (Sonthemeier and Ransom, Patch Clamp Analysis Advanced Techniques, Human Press, 2002:35-67). As shown in the picture (right), differentiated podocytes are very large and flattened and have a high membrane capacitance (90~200 pF for podocytes vs ~12 pF for HEK293 cells). This feature seriously interferes with voltage clamping across the whole cell membrane. In order to reduce space-clamp errors, we recorded the current within 2 h after cell seeding to maintain a relatively small and spherical cell shape. Under the whole-cell dialyzed patch-clamp configuration, ER Ca^{2+} depletion triggered CRAC currents in differentiated mouse podocytes, showing characteristic inwardly rectification (see new Fig. 1f).

For the accurate analysis of the intrinsic channel kinetics of CRAC currents regulated by insulin, we used HEK293FT cells expressing mCherry-Flag-tagged Orai1 and YFP-STIM1 (0.2 μg each per 35-mm dish) that are much smaller in size (mean membrane capacitance: approximately 12 pF). As mentioned by the reviewer, Staruschenko's group recorded single-channel TRPC6 currents but not whole-cell current in isolated glomeruli (Ilatovskaya et al., Am J Soc Nephrol, 29(7):1919-27, 2018; Ilatovskaya et al., Sci Rep, 7:299, 2017; Ilatovskaya et al., Kid Int, 86(3):506-14, 2014). Our lab has extensive experience with electrophysiology experiments, including single-channel analysis of TRPC6 and TRPV5 (An et al., Curr Biol, 21(23):1979-87, 2011; Cha et al., J Memb Biol, 220(1-3):79-85, 2007). The conductance and current amplitude of TRPC6 is 35 ± 3 pS in symmetrical 140 mM Na^+ and ~ 2.3 pA at -80 mV, respectively (An et al., Curr Biol, 21(23):1979-87, 2011). The distinctive characteristic of the CRAC channel is its extremely small unitary conductance in 10~25 fS, estimated from noise analysis (Krizova et al., Eur Biophys J, 48(5):425-45, 2019; Prakriya and Lewis, J Gen Physiol, 119:487-507, 2002; Zweifach and Lewis, Proc Natl Acad Sci USA, 90:6295-99, 1993). Due to this attribute, single-channel CRAC current recordings were unfeasible up to now (Krizova et al., Eur Biophys J, 48(5):425-45, 2019; Prakriya and Lewis, Cell Calcium 33:311-21, 2003;

Prakriya and Lewis, *J Gen Physiol*, 128:373-86, 2006). This tiny unitary conductance probably underlies the very narrow pore size of this channel compared with that of other Ca²⁺-selective channels; for example, the full- and sub-conductance values of highly Ca²⁺-selective TRPV5 channel were 59±6 and 29±3 pS, respectively (Cha et al., *J Memb Biol*, 220(1-3):79-85, 2007). Staruschenko's group also evaluated TRPC6 activity through Ca²⁺-imaging using Fluo4 in isolated glomeruli (Golosova et al., *Life Sci Alliance*, 3(12):e202000853, 2020; Ilatovskaya et al., *Am J Soc Nephrol*, 29(7):1919-27, 2018; Ilatovskaya et al., *Sci Rep*, 7:299, 2017; Ilatovskaya et al., *Kid Int*, 86(3):506-14, 2014).

To date, we have not overcome the technical difficulties in whole-cell voltage clamping or single channel recording from isolated glomeruli. However, our findings from live-cell Ca²⁺ imaging in isolated glomeruli from control or *Orai1* deletion mice and electrophysiologic studies using *Orai1*-overexpressed HEK293FT cells could demonstrate insulin-activated SOCE via *Orai1* upregulation, a potentially critical process in actin cytoskeleton remodeling and impairment of the glomerular filtration barrier.

2. The requirement to show that loss of function of *Orai* is protective just as gain of function (transgenics/overexpression) causes disease is also a requirement for publication of this manuscript. The reason is again critically important and simple: the transgenic overexpression of a calcium permeable channel can cause cell injury/death because of calcium overload independent of any signaling events or specific events related to the biology of this cell, meaning the overexpression experiments on which this manuscript relies are greatly prone to artifacts and can mislead the investigators to erroneous conclusions. For example, if one were to overexpress a voltage gated calcium channel in a podocyte, that may also lead to cell injury and proteinuria due to calcium overload/podocyte death, but there is no evidence that voltage gated calcium channels are meaningfully involved in the biology of this cell type. Therefore, the only way to control and secure against this possible artifact is to perform loss of function studies to show that podocyte specific *Orai* knockout mice are protected from podocyte injury/proteinuria (as caused by a number of agents such as protamine sulfate, puromycin aminonucleoside, Adriamycin, high glucose diet etc). This must be done for the manuscript to be credible.

Our response:

We agree with the point raised by the reviewer that the overexpression of the transgenic Ca²⁺-permeable channel alone can cause cell injury or death because of Ca²⁺ overload. Based on the reviewer's suggestion of conducting a loss-of-function study, we generated podocyte-specific *Orai1*-deletion mice by crossing floxed *Orai1* mice with podocyte-specific Cre recombinase mice driven by the podocin promoter/enhancer region (see new Supplementary Fig. 3a-b).

We confirmed *Orai1* deletion from its reduced mRNA and protein levels in the glomeruli isolated from podocyte-specific *Orai1*-deletion (*Cre+;fl/fl*) mice compared with those in glomeruli from the wild-type or heterozygous (*Cre+;fl/+*) mice (see new Fig. 3a, b). Immunofluorescence staining for a podocyte-specific marker, synaptopodin, demonstrated the selective reduction of *Orai1* in the podocytes of *Cre+;fl/fl* mice (see new Fig. 3c).

Podocyte-specific *Orai1*-deletion mice were viable and demonstrated no apparent changes in kidney size and renal histology (see new Supplementary Fig. 3c and new Fig. 3d). As expected, native SOCE in isolated glomeruli was significantly reduced in podocyte-specific *Orai1*-deletion mice compared with that in wild-type or heterozygous mice (new Fig. 3e and f), supporting the notion that *Orai1* is the main component of SOCE in podocytes *in vivo*.

Functionally, we demonstrated the exclusive role of *Orai1* in the podocyte actin remodeling and proteinuria triggered by insulin. Podocyte-specific *Orai1*-deletion abrogated insulin-activated SOCE and synaptopodin dissolution in isolated glomeruli and albuminuria *in vivo* (see new Fig. 3g,h,i,j,k). These data strongly support that podocyte *Orai1* is a crucial target for insulin-stimulated detrimental effects on the glomerular filtration barrier.

3. Consideration of the effects of other TRPC channels was also raised as a major concern by this reviewer. The authors have done studies to show that *Orai* knockdown does not alter the expression of TRPC5 and TRPC6 channels, but since ion channel activity rather than expression is what is critically important here, we return to point 1, which is that electrophysiology studies in podocytes looking at whether TRPC5, TRPC6 or other TRPC or *Orai* currents are affected by detailed pharmacology and genetic studies (using all available drugs and using knockdowns for each of those channels) is the only way to separate between these different channels, and support the novelty of the findings in this manuscript.

The fact that insulin mediates calcium influx in TRPC6 knockout mice, which is offered as a new study in support of the conclusion that this is due to *Orai* channel activity is not conclusive because it does not address any possible contributions from TRPC5 or other calcium permeable channels and it relies on drugs that are not specific for *Orai*.

Our response:

We appreciate the reviewer's constructive suggestion, which allowed us to clarify and highlight the impact of this study. As suggested by the reviewer, we have performed further experiments to demonstrate the exclusive role of *Orai1* and to support the novelty of the findings.

Firstly, in additional experiments using *db/m* and *db/db* mice, we observed that *Orai1* upregulation was dominant at the early stage (11 wk), coinciding with hyperinsulinemia, while TRPC6 expression increased at the late stage (19 wk) in glomeruli from *db/db* mice (see new Fig. 5e, f). Immunofluorescence images also showed higher *Orai1* expression in glomerular cells obtained from 11-week-old *db/db* mice than in those obtained from *db/m* control mice (new Fig. 5g). As a downstream effector of the Ca^{2+} /calcineurin pathway, synaptopodin levels

remained decreased at both 11 and 19 weeks of age in *db/db* mice, suggesting that Ca^{2+} -triggered podocyte injury is maintained by the action of TRPC6 after Orai1 decline *in vivo* (new Fig. 5h, i).

Secondly, as described in the response to comment #2, podocyte-specific *Orai1*-deletion abrogated insulin-activated SOCE and synaptopodin degradation in isolated glomeruli and albuminuria (see new Fig. 3g,h,i,j,k), supporting that podocyte Orai1 is a crucial target for insulin's detrimental actions on the glomerular barrier. Conversely, genetic ablation of TRPC6 (*Trpc6*^{-/-}) had a minor effect on insulin-induced albuminuria in mice (new Supplementary Fig.4) as well as SOCE (Supplementary Fig.1). These findings strongly suggest that a higher level of insulin induces albuminuria by targeting Orai1 rather than TRPC6 in podocytes.

As in the response to comment#1 from reviewer#1 and to comment#1 from reviewer#3, in the previous version of manuscript, it was reported that insulin stimulates TRPC6, not TRPC5, by increasing the cell surface abundance of TRPC6 in a NADPH oxidase-dependent mechanism in cultured podocytes, suggesting that both channels are differentially regulated by growth factors (Kim et al., *Am J Physiol Renal Physiol*, 302:F298-307, 2012). Similarly, high glucose, another factor from the diabetic milieu, increased TRPC6 expression, but not TRPC5, as well as reducing the expression of podocin and nephrin in human podocytes *in vitro* (Sonneveld et al., *Am J Pathol*, 184:1715-26, 2014). On the other hand, high glucose decreased Orai1 protein expression in rat mesangial cells and promoted Orai1-induced Ca^{2+} entry in cardiomyocytes (Jiang et al., *Am J Physiol Renal Physiol*, 314:F855-863, 2018; Wu et al, *Cell Death Dis*, 12(2):216, 2021). In our study, high glucose had no effect on Orai1-mediated SOCE in cultured mouse podocytes (right). Our data from previous studies demonstrated that Orai1 and TRPC6/5 are differentially regulated by the diabetic milieu such as hyperinsulinemia and hyperglycemia.

Regarding the other aspect, as described in the Discussion section of revised manuscript, multiple studies have provided compelling evidence that $[\text{Ca}^{2+}]_i$ links actin dynamics to podocyte dysfunction and proteinuria. Among Ca^{2+} -permeable ion channels, TRPC5 and TRPC6 channels play a crucial role in $[\text{Ca}^{2+}]_i$ regulation and the actin dynamics of podocytes, especially, antagonistic regulation of actin dynamics and cell motility (Greka and Mundel, *J Am Soc Nephrol*, 22(11):1969-80). Under physiological conditions, angiotensin II receptor stimulation activates both TRPC5 and TRPC6, which have different phenotypes associated with actin dynamics and cellular behaviors. TRPC5-mediated Ca^{2+} influx decreases stress fiber assembly and promotes cell migration, whereas TRPC6 activation promotes stress fiber formation and a contractile cell phenotype (Greka and Mundel, *J Am Soc Nephrol*, 22(11):1969-80, 2011; Tian et al., *Sci Signal*, 3(145):ra77, 2010). In our study, however, Orai1-mediated Ca^{2+} influx decreased stress fiber formation and increased contractility, demonstrating the distinct role of Orai1 in actin dynamics compared with that of TRPC5 or TRPC6 in podocytes. We are going to dissect the Ca^{2+} microdomain and the downstream

effectors of Orai1 and TRPC5/6 channels for linking the actin dynamics to podocytopathy under diabetic milieus in a future investigation.

Additional major ongoing concerns:

4. The pharmacologic studies (as in my previous point 6) remain problematic as the drugs used are non-specific and can have vast effects on the actin cytoskeleton. It is impossible to separate cause and effect here, despite the new studies (with a focus on VAMP2), which however have the same limitations because they rely on the same non-specific drugs. Genetic studies and electrophysiology to bolster these conclusions would be required for this part of the manuscript to be convincing. For example, since there are major concerns about the surface trafficking studies, it would be convincing to record Orai channel activity in podocytes under these conditions to conclusively demonstrate that Orai channels are absent or present in the plasma membrane. In the absence of this, the focus on exocytosis/trafficking as a mechanism for the regulation of Orai is not conclusive and contradicts previously reported work, as also noted by another reviewer.

Our response:

Yes, we agree with your comments about the limitation of pharmacologic studies such as non-specific and/or pleiotropic effects of brefeldin A (BFA) and BoNT. In the previous version of the manuscript, we carried out additional pharmacological experiments using tetanus toxin (TeNT), which cleave v-SNARE protein VAMP2 to inhibit vesicular exocytosis. The disruption of the SNARE complex by cleaving VAMP2 using tetanus toxin (TeNT) impairs SOCE (Rosado et al., *J Cell Physiol*, 205:262-9, 2005; Alderton et al., *Cell Calcium*, 28:161-9, 2000). We have previously reported that serum growth factors stimulate PI3K-dependent TeNT-sensitive exocytosis of TRPC6 and Orai1 in podocytes and HEK293 cells, respectively (Kim et al., *Am J Soc Nephrol* 28(1):140-51, 2017; Kim et al., *Pflueger's Arch* 473(4):647-58, 2021). In this revised manuscript, we report that TeNT decreased the insulin-promoted surface abundance of Orai1 in podocytes, supporting that insulin increases VAMP-2 associated exocytosis of Orai1 (Fig. 3e,f in previous MS, Supplementary Fig. 6d in current M). In addition to pharmacologic studies using BFA or TeNT, VAMP2 knockdown also attenuated the insulin-induced increase of SOCE and cell-surface abundance of Orai1, supporting the notion that insulin promotes SOCE via exocytosis of the channel (revised Supplementary Fig. 6a-d and Fig. 4g-j). Furthermore, we confirmed that VAMP2 physically interacted with Orai1 as well as TRPC6 or TRPC3 (Singh et al., *Mol Cell*, 15(4):635-46, 2004).

We and others reported that PI3K-dependent signaling contributes to the stimulation of exocytosis of multiple channels such as TRPC5, TRPC6, and Orai1 (Kim et al., *Pflueger's Arch* 473(4):647-58, 2021; Dalton et al., *Proc Natl Acad Sci USA*, 114:752-57, 2017; Kim et al., *Am J Soc Nephrol* 28(1):140-51, 2017; Xie et al., *Nat Commun*, 3:1238, 2012; Bezzerides et al., *Nat Cell Biol* 6:709-20, 2004). We also showed that PI3K inhibitors prevented insulin-stimulated SOCE and decreased the cell-surface abundance of Orai1 in mouse podocytes (Fig. 4 k, l,m,n). Taken together, these findings suggest that the insulin-stimulated PI3K

signaling cascade may be a common regulator of the exocytic pathway for Orai1 and TRPC6, at least in part, in podocytes.

5. The authors point out correctly that endogenous Orai is not only detected in podocytes but also in mesangial cells and tubular cells (Fig. 1 and new Fig 7d as well) (i.e. it is present throughout the entire kidney). Since insulin can also act on all these different cell types in the kidney, how can we be sure of the specificity of the insulin-Orai pathway in podocytes as claimed in this study? Again, the loss of function experiment is critical here: the podocyte specific deletion of Orai would be the only way to be convincing, if one can show that the deletion of Orai in podocytes is protective of podocyte injury/proteinuria.

Our response:

We agree with the reviewer that endogenous Orai1 is not only detected in podocytes but also in mesangial cells and tubular cells. To validate the specificity of the insulin–Orai1 pathway in podocytes, we carried out additional experiments using podocyte-specific Orai1 deletion mice, as described in the response to comment #2. We confirmed podocyte-specific deletion of Orai1 by immunofluorescence staining, quantitative real-time PCR, western blot, and SOCE measurement in isolated glomeruli from Cre+; *fl/fl* mice (see new Fig. 3a-f, Supplementary Fig. 3). Augmented SOCE, albuminuria, and synaptopodin degradation following insulin treatment were abrogated by podocyte-specific *Orai1*-deletion (see new Fig. 3g,h,i,j,k). These loss-of-function experiments support the specificity of the insulin–Orai1 signaling nexus in podocyte injury and proteinuria.

6. The new data in Fig. 7 (to address point 9 from previous review) are not sufficient to support the conclusion of this study in db/db mice. Again, only calcium imaging is employed to address what the authors call SOCE (store operate calcium entry) but these studies can not demonstrate that this is specifically related to Orai channel activity. Again, patch clamp electrophysiology in intact db/db mouse glomeruli and in podocyte-specific Orai knockout crossed to db/db mice is critically important to demonstrate this point here.

Our response:

As explained in the response to comment #1, we could not record single-channel or whole-cell CRAC currents in isolated glomeruli due to the extremely small unitary conductance or space-clamp errors, respectively (Krizova et al., Eur Biophys J, 48(5):425-45, 2019; Prakriya and Lewis, Cell Calcium 33:311-21, 2003; Prakriya and Lewis, J Gen Physiol, 128:373-86, 2006). We agree that SOCE measurement using live-cell Ca²⁺ imaging is not specific to Orai channel activities. Therefore, to validate Orai1 as a target for insulin, we applied the loss-of-function strategy using mice with genetic deletion of Orai1. First, we confirmed that SOCE in was markedly decreased in the glomeruli isolated from podocyte-specific Orai1 deletion mice, implying that Orai1 in podocytes is mainly responsible for SOCE in isolated glomeruli (see Fig.

3e,f,g,h). Second, we showed that an Orai1 inhibitor GSK7975A abolished SOCE in glomeruli isolated from db/db mice (see Fig. 6a and b). Our genetic and pharmacologic studies support the notion that Orai1 in podocytes is an essential component of store-operated Ca^{2+} entry *in vivo*.

We also agree with reviewer's suggestion that experiments using podocyte-specific Orai1 knockout mice crossed to db/db mice can strongly support the conclusion of our study. However, there is a practical difficulty that db/db mice are genetically infertile (Garris and Garris, *Exp Biol Med* (Maywood), 228(9):1040-50, 2003). Therefore, in order to get db/db mice, we would need to cross from db/m mice. Theoretically, when we cross the db/m and podocin-Cre Orai1 floxed mice, the probability of obtaining db/m- podocin-cre Orai1 f/+ mouse is 1/4. A subsequent cross with the db/m-podocin Cre Orai1 f/+ would yield db/m-podocin Cre Orai1 f/f mice, with a probability of 1/8. As a consequence, the last crossing ratio maybe 1/4. If podocin-Cre Orai1 mice are ready for crossing and all mice are able to reproduce, the probability of obtaining podocyte-specific Orai1 knockout db/db mice becomes 1/128. In fact, we were eager to generate podocyte-specific Orai1 deletion in db/db mice; however, the practical ratios were not the same as the theoretical ratio. Although we calculated the probability of obtaining db/m-podocin-Cre Orai1 f/+ mouse as 1/4, the real pups' genotypes were not as estimated. In our experience, the ratio of db/m-podocin-Cre Orai1 f/+ mouse was much less than 1/4. Unfortunately, we could not obtain these mice to conduct the experiments.

Reviewer #2 in the comments to our previous version of manuscript recommended the use of GSK7975A, a pyrazole derivative, as a selective inhibitor for Orai1 and Orai3 channels (Derler et al., *Cell Calcium* 53:139-51, 2013). Following the reviewer's suggestion, we have carried out additional studies using GSK7975A in isolated glomeruli and *in vivo* experiments with db/db mice. All pathological changes in db/db mice, including Orai1 upregulation, synaptopodin dissolution, slit diaphragm disruption, and albuminuria, were restored following administration of GSK7975A, similar to the results obtained following the treatment with 2-APB or SKF96365. While 2-APB at high concentration inhibits Orai1 channel in a STIM1-dependent manner, 2-APB directly activates Orai3 channel which is independent of ER Ca^{2+} depletion and STIM1 (Prakriya and Lewis, *Physiol Rev.* 95:1383-436, 2015). Taken together, the results from the repertoire of blockers used in our study support the notion that Orai1, as the main component of SOCE, participates in the impairment of the glomerular filtration barrier in db/db mice.

7. In response to point 10 from previous review, the authors have not performed any genetic studies as suggested, and instead have relied again on drugs that they have already acknowledged are non-specific for Orai or even for SOCE. The only convincing way to address the issue of lack of drug specificity is to use Orai knockout mice.

Our response:

As described in the response to comment #2, we generated podocyte-specific *Orai1*-deletion mice by crossing floxed *Orai1* mice with podocyte-specific Cre recombinase mice driven by the podocin promoter/enhancer region (new Supplementary Fig. 3a-b) to ensure specific insulin targeting of podocyte *Orai1*. We validated *Orai1*-induced native SOCE in isolated glomeruli using podocyte-specific *Orai1*-deletion mice (see new Fig. 3e and f). We also demonstrated the exclusive role of *Orai1* in insulin-triggered SOCE activation, synaptopodin depletion, and albuminuria *in vivo*. Podocyte-specific *Orai1*-deletion abrogated the insulin-activated SOCE in isolated glomeruli, albuminuria, and synaptopodin degradation (see new Fig. 3g,h,i,j,k). The results of genetic studies conducted according to the reviewer's guidance strongly support that podocyte *Orai1* is a crucial target for insulin-induced detrimental effects on the glomerular filtration barrier.

8. In response to point 11 from previous review, the authors suggest that the short time scale of their observations is sufficient to support the conclusion that that NFAT-mediated transcriptional regulation of *Orai* (or other channels as previously published for TRPC6) is not relevant in their studies. This is an erroneous conclusion, as transcriptional events are well known to occur on short time scales.

Further, rescue of synaptopodin degradation by calcineurin has been previously shown to be the result of inhibiting the effect of Cathepsin L in podocytes (Faul et al, Nat Med, 2008) and more recently as the result of blocking Vav2 activity in podocytes (Buvall et al, JASN, 2017). None of these pathways (as well as the mitochondrial pathway) require or implicate *Orai* activity and therefore the response offered to this point does not actually address the point.

Our response:

We appreciate the reviewer's comments. We agree that transcriptional events can be regulated over a short time scale. In the revised manuscript, we have clarified that insulin transiently increased *Orai1*-mediated Ca^{2+} influx followed by podocyte actin remodeling and synaptopodin dissolution. The SOCE activation by insulin was detected after 10 min incubation and peaked at 1 h, without any increased in *Orai1* protein levels (see Fig. 4d and 4e). Moreover, in the former version of manuscript, the mRNA expression of *Orai1* was not significantly upregulated in the *db/db* mice, implying the increased stability of *Orai1* protein and not increased transcription via Ca^{2+} -calcineurin-NFAT signaling.

As summarized in Figure 7, acute (or short-term) stimulation by insulin promotes *Orai1*-mediated Ca^{2+} entry by increasing the cell-surface abundance of the *Orai1* channel without increasing its mRNA and protein levels. This event causes reversible physiological changes in the podocyte actin cytoskeleton, providing slit diaphragm plasticity and/or resilience against injury. Chronic *Orai1* stimulation by insulin also accelerates *Orai1*-mediated SOCE by increasing *Orai1* protein stability rather than transcription, thereby leading to irreversible pathological changes such as podocyte foot process effacement and severe proteinuria.

Yes, the rescue of synaptopodin degradation by calcineurin has been previously demonstrated to be the result of inhibiting the effect of Cathepsin L in podocytes (Faul et al, Nat Med, 2008) and, more recently, as the result of blocking Vav2 activity in podocytes (Buvall et al, JASN, 2017). Our study focused on insulin–Orai1 signaling acting on synaptopodin-dependent actin remodeling. Calcineurin dephosphorylates the 14-3-3 binding site in synaptopodin, promoting its degradation by cathepsin L and Vav2 activation. In addition to Cathepsin L and Vav2 as downstream effectors of calcineurin, there are multiple upstream regulators and downstream effectors of Ca^{2+} /calcineurin. For example, calcineurin also modulates podocyte apoptosis by regulating Drp1- or BAD or is cleaved by Ca^{2+} -dependent protease Calpain in podocyte injury models (Spurney, Front Endocrinol, 5, 181, 2014; Ding et al, PLoS One, 2016). In our future study, we will dissect the role of upstream and/or downstream effectors of synaptopodin as a coincidence detector upon Orai1-calcineurin pathway.

Multiple studies have demonstrated that mitochondria are key regulators of Orai1 activity. CsA can block the opening of mitochondrial permeability transition pore (PTP) in response to pathological mitochondrial Ca^{2+} overload and prevent podocyte apoptosis via Drp1- or Bad-dependent mitochondrial apoptosis pathways. Apoptosis prevention could be another beneficial action of CsA against glomerular diseases. The link between Orai1-mediated Ca^{2+} signaling and mitochondrial function merits further study.

Unfortunately, overall the revised manuscript does not address the major concerns raised and therefore it is not convincing as to the role of Orai in podocyte pathobiology.

REVIEWERS' COMMENTS

Reviewer #3 (Remarks to the Author):

All comments have been addressed very nicely